



# Top of the Atmosphere Reflected Shortwave Radiative Fluxes from GOES-R

Rachel T. Pinker[1], Yingtao Ma[1], Wen. Chen[1], Istvan Laszlo[2], Hongqing Liu[3],
Hye-Yun Kim[3] and Jamie Daniels[2]
[1] Department of Atmospheric and Oceanic Science, University of Maryland, College Park, MD
[2] NOAA NESDIS Center for Satellite Applications and Research, College Park, MD
[3] I.M. Systems Group, Inc., Rockville, MD
Correspondence to: Rachel T. Pinker (pinker@atmos.umd.edu)
**Abstract.** Under the GOES-R activity, new algorithms are being developed at the National Oceanic and
Atmospheric Administration (NOAA)/Center for Satellite Applications and Research (STAR) to derive
surface and Top of the Atmosphere (TOA) shortwave (SW) radiative fluxes from the Advanced Baseline
Imager (ABI), the primary instrument on GOES-R. This paper describes a support effort in the
development and evaluation of the ABI instrument capabilities to derive such fluxes. Specifically, scene
dependent narrow-to-broadband (NTB) transformations are developed to facilitate the use of observations
from ABI at the TOA. Simulations of NTB transformations have been performed with MODTRAN4.3
using an updated selection of atmospheric profiles as implemented with the final ABI specifications.
These are combined with Angular Distribution Models (ADMs), which are a synergy of ADMs from the
Clouds and the Earth's Radiant Energy System (CERES) and from simulations. Surface condition at the
scale of the ABI products as needed to compute the TOA radiative fluxes come from the International
Geosphere-Biosphere Programme (IGBP). Land classification at 1/6º resolution for 18 surface types are
converted to the ABI 2-km grid over the (CONtiguous States of the United States) (CONUS) and
subsequently re-grouped to 12 IGBP types to match the classification of the CERES ADMs. In the



simulations, default information on aerosols and clouds is based on the ones used in MODTRAN.
Comparison of derived fluxes at the TOA is made with those from the CERES and/or the Fast Longwave
and Shortwave Radiative Flux (FLASHFlux) data. A satisfactory agreement between the fluxes was
observed and possible reasons for differences have been identified; the agreement of the fluxes at the
TOA for predominantly clear sky conditions was found to be better than for cloudy sky due to possible
time shift in observation times between the two observing systems that might have affected the position
of the clouds during such periods.
**1 Introduction**
When a new satellite is contemplated, the exact characteristics of the newly selected sensors are not fully
known; simulations of proposed sensors are also not readily available. Yet, there is a need to obtain a
priori information on the expected performance of the new instruments. This is usually accomplished by
using characteristics of instruments in closest resemblance to the proposed ones and performing
simulations that can provide insight on the expected performance of the new instrument. As such, an
evolutionary process can be expected and it did precede activities reported in this manuscript.
The "indirect path method" used at the Center for Satellite Applications and Research (STAR) (Laszlo et
al., 2020) for deriving SW radiative fluxes from satellite observations requires knowledge of the SW
broadband (0.2 – 4.0 µm) top of the atmosphere (TOA) albedo. The Advanced Baseline Imager (ABI)
observations onboard of the NOAA GOES-R series of satellites provide reflectances in six narrow bands
in the shortwave spectrum (**Table 1**); these must be first transformed into broadband reflectance (the
narrow-to-broadband, NTB, conversion process), and then the broadband reflectance must be transformed
into a broadband albedo (the ADM conversion process).
During the pre-launch activity NTB transformations were developed based on theoretical radiative
transfer simulations with MODTRAN-3.7 and 14 land use classifications from the International
Geosphere-Biosphere Programme (*IGBP*) (Hansen et al., 2010). They were augmented with ADMs from



(CERES) observed ADMs (Loeb et al., 2003) and theoretical simulations (Niu and Pinker, 2011) to
compute TOA fluxes. The resulting NTB transformations and ADMs have been tested using proxy data
and simulated ABI data. The proxy instruments used in the simulations include the GOES-8 satellite, the
Advanced Very-High Resolution Radiometer (AVHRR) sensor on the Polar Orbiting satellites, the
Spinning Enhanced Visible Infra-Red Imager (SEVIRI) sensor on the European METEOSAT Second
Generation (MSG) satellites, and the Moderate Resolution Imaging Spectroradiometer (MODIS)
instrument on the NASA Terra and Aqua Polar Orbiting satellites (Pinker et al*.,* 2021, unpublished). For
each of these satellites, the evaluation of the methodologies was done differently; some results were
evaluated against ground observations while others, against TOA information from CERES as well as
from the (ESA) Geostationary Earth Radiation Budget (GERB) satellite (Harries et al., 2005). The results
obtained provided an insight on the expected performance of the new ABI sensor. Those procedures have
been subsequently updated and applied to the new ABI instrument once it was built and fully
characterized.
In this paper we describe activity in support of methodologies to derive surface shortwave (SW) radiative
fluxes from the operational Advanced Baseline Imager (ABI) instrument on the GOES-R series of the
NOAA geostationary meteorological satellites. We describe the physical basis and the development of
the (NTB) transformations of satellite observed radiances and the bi-directional corrections to be applied
to the broadband reflectance to obtain broadband TOA albedo. The methodology will be presented in
section 2, results in section 3 and a summary and discussion in section 4.

**2. Methodology**

The following two flowcharts (**Figs. 1 and 2**) describe the necessary steps to derive the NTB
transformations and the ADMs. Details of these two steps will follow.



The TOA narrowband and broadband reflectances can be calculated from the spectral radiances
simulated from MODTRAN 4.3 and the response functions of the satellite sensor as shown in equations
(1) and (2):
$$\rho_{nb}(\theta_0,\theta,\phi) = \frac{\pi \int\limits_{\lambda 1}^{\lambda 2} I(\lambda,\theta_0,\theta,\phi)G(\lambda)d\lambda}{\int\limits_{\lambda 1}^{\lambda 2} \cos(\theta_0)S_0(\lambda)G(\lambda)d\lambda} \qquad (1)$$

$$\rho_{bb}(\theta_0,\theta,\phi) = \frac{\pi \int\limits_{0.2\,\mu m}^{4\,\mu m} I(\lambda,\theta_0,\theta,\phi)d\lambda}{\int\limits_{0.2\,\mu m}^{4\,\mu m} \cos(\theta_0)S_0(\lambda)d\lambda} \qquad (2)$$


where $\rho_{nb}$ is narrowband reflectance; $\rho_{bb}$ is broadband reflectance; $\theta_0$: solar zenith angle; $\theta$: view
(satellite) zenith angle; $\phi$: relative azimuth angle;
$I_\lambda$: reflected spectral radiance; $S_0(\lambda)$: solar spectral irradiance;
$G_\lambda$: spectral response functions of satellite sensors; $\lambda_1$ and $\lambda_2$ are the spectral limits of the sensor spectral
band.
As stated previously, the ADMs from CERES-based observations (Loeb et al., 2003) were augmented
with theoretical simulations (Niu and Pinker, 2011) to compute TOA fluxes. This due to the fact that
CERES observations at higher latitudes are under-sampled or not existent.
The combined ADMs are developed for each angular bin by weighting the modeled and CERES ADMs
based on the number of samples used to derive the ADMs of each type (Niu et al., 2011). Specifically:
$$\overline{R}(\theta_0,\theta,\phi) = \frac{1}{m+n}\big(m \times R_{CERES}(\theta_0,\theta,\phi) + n \times R_S(\theta_0,\theta,\phi)\big) \qquad (3)$$

$\overline{R}(\theta_0,\theta,\phi)$:      averaged ADMs at each angular bin;



| 94 | $R_{CERES}$: | anisotropic factor from CERES ADMs; |
|---|---|---|
| 95 | $R_S$: | anisotropic factor from simulated ADMs; |
| 96 | $m$ and $n$: | observation numbers at angular bins for CERES and simulated ADMs. |



**2.1 Selection of Atmospheric profiles for simulations**


We have selected 100 atmospheric profiles covering the globe and the seasons, to use as input for
simulations with MODTRAN4.3. A tool was developed to select profiles from a Training Data set known
as SeeBor Version 5.0 (https://cimss.ssec.wisc.edu/training_data/) (Borbas et.al. 2005). Originally it
consisted of 15704 global profiles of temperature, moisture, and ozone at 101 pressure levels for clear
sky conditions. The profiles are taken from NOAA-88, and the European Centre for Medium-Range
Weather Forecasts (ECMWF) 60L training set, TIGR-3, ozone-sondes from 8 NOAA Climate Monitoring
and Diagnostics Laboratory (CMDL) sites, and radiosondes from the Sahara Desert during 2004. A
technique to extend the temperature, moisture, and ozone profiles above the level of existing data was
also implemented by the providers (University of Wisconsin-Madison, Space Science and Engineering
Center, Cooperative Institute for Meteorological Satellite Studies (CIMSS). **Fig. 3** shows the selected
profile locations; each season includes 25 profiles.
The SeeBor profiles are clear sky profiles. The top of the profiles is at 0.005 mb which is about 82.6 km.
We did an experiment to check the impact of reducing the number of levels for a profile (initially,
we have used only 40 levels). In the experiment computed were radiances from profiles with 50
levels as well as radiances from profiles with 98 Levels. The difference between the two radiances
(50 lev-98 lev) were below 5 % reaching 15 % around 2.5 µm. In the experiment we used the odd
number levels starting from surface (plus the highest level) to reduce the number of profile levels.
Based on these experiments we have opted to keep all 98 profile levels.



The atmospheric profiles at each pressure level include temperature, water vapor and ozone. The surface
variables include surface skin temperature, 2 m temperature, land/sea mask, and albedo. We have
conducted a thorough investigation how the selected profiles represent the entire sample of 15704 profiles.
An example showing the comparison of temperature, humidity and ozone profiles is shown in **Fig. 4.** As
seen, there is a positive bias in the selected profiles due to their higher concentration at the lower latitudes.
Since our domain of study is in such latitudes this selection should not have adverse effects on the
simulations.
**2.2 Surface conditions**

Surface condition is one of the primary inputs into the MODTRAN simulations. The International
Geosphere-Biosphere Programme (IGBP) land classification is used as data source (Hansen et al., 2010;
Loveland et al., 2010). The dataset is at 1/6-degree resolution and includes 18 surface types. We have
converted the 1/6º (~18.5 km) resolution to the ABI 2-km grid using the nearest grid method **(Fig. 5)**. The
method for cloudy sky uses 4 surface types; these are also derived from 12 IGBP types (**Table 2**).

**2.3 Clear and cloudy sky simulations**

Under clear sky, multiple scattering from aerosols is important. We have included 6 aerosol types (**Table
3**) to cover a range of possible conditions under clear sky. Aerosol models are selected based on the type
of extinction and a default meteorological range for the boundary-layer aerosol models as listed below:
Aerosol Type 1: Rural extinction, visibility = 23 km
Aerosol Type 4: Maritime extinction, visibility = 23 km
Aerosol Type 5: Urban extinction, visibility = 5 km
Aerosol Type 6: Tropospheric extinction, visibility = 50 km
Aerosol Type 8: Advective Fog extinction, visibility = 0.2 km
Aerosol Type 10: Desert extinction, visibility based on wind speed





For the 6 aerosol types, the total number of MODTRAN simulations for each surface type is 288,000.
When doing NTB simulation, we use all 6 types of aerosols. The Rural, Ocean, Urban and Fog aerosols
are distributed in the lower 0-2 km region. Tropospheric aerosol is distributed from 0 to 10 km tropopause.
The Rural, Ocean, Urban and Tropospheric aerosol optical properties have Relative Humidity (RH)
dependency. The Single Scattering Albedo (SSA) is given on 4 RH grids (0, 70, 80, 99) on a spectral grid
of 788 points ranging from 0.2 to 300 microns. The Desert aerosol is wind speed dependent and the optical
properties are given for 4 wind speeds (0, 10, 20, 30).
Simulations were performed for ABI for all the cloud cases described in **Table 3.** To merge cloud layers
with atmospheric profiles we have followed the procedure as described in *Berk et al.* (1985, 1998),
namely: "Cloud profiles are merged with the other atmospheric profiles (pressure, temperature, molecular
constituent, and aerosol) by combining and/or adding new layer boundaries. Any cloud layer boundary
within half a meter of an atmospheric boundary layer is translated to make the layer altitudes coincide;
new atmospheric layer boundaries are defined to accommodate the additional cloud layer boundaries."
100% relative humidity is assumed within the cloud layers (default).

**2.4 Selection of angles**

The total number of angles used in the simulations is given in **Table 4**. The selected spectral grids for
solar zenith angles, satellite view angles and azimuth angles are at Gaussian quadrature points, plus 0º to
solar zenith angles (sza) and satellite viewing angles (vza) and 0º and 180º (forward and backward view)
to the satellite relative azimuth angles. Solar angle and satellite view angle are referenced to target or
surface for satellite simulation with 0º meaning looking up (zenith). Azimuth angle is defined as when
the relative azimuth angle equals 180º, the sun is in front of observer.
The definitions of solar zenith angle and azimuth angle in this table corresponds to the definitions of
MODTRAN but that is not the case for the satellite zenith angle. MODTRAN uses nadir angle as 180º-
satellite zenith angle, ignoring spherical geometry.




## 2.5 Selection of optimal computational scheme


Computational speed is an issue for simulations that account for multiple scattering. MODTRAN4.3
provides three multiple scattering models (Isaacs, DISORT, and Scaled Isaacs) and three band models at
resolutions (1 cm$^{-1}$, 5 cm$^{-1}$, and 15 cm$^{-1}$). The DISORT model (Stamnes et al., 1988) provides the most
accurate radiance simulations but the runs are very time consuming. The Isaacs (Isaacs et al. 1987) 2-
stream algorithm is fast but oversimplified. The Scaled Isaacs method performs radiance calculations at
a small number of atmospheric window wavelengths. The multiple scattering contributions for each
method are identified and ratios of the DISORT and Isaacs methods are computed. This ratio is
interpolated over the full wavelength range, and finally, applied as a multiple scattering scale factor in a
spectral radiance calculation performed with the Isaacs method.
To optimize simulation speed and accuracy, we performed various sensitivity tests, including
combinations of multiple scattering models, band resolution, and number of streams. **Table 5** lists
simulation options and their corresponding calculation speed. The most computationally extensive option
is DISORT 8-stream with 1 cm$^{-1}$ resolution which requires 930 seconds to finish one single run. The
fastest is Scaled Isaacs with 15 cm$^{-1}$ resolution which only needs 6.67 seconds. Number of streams does
not affect the Scaled Isaacs calculation speed. This is different from Isaacs and DISORT for which both
stream number and band resolution have notable effects.
Based on results presented in **Table 5**, the efficient options (< 40 seconds) are Isaacs, DISORT 2-stream
with 15 cm$^{-1}$, DISORT 4-stream 15 cm$^{-1}$, and Scaled Isaacs all streams at all resolutions. Although the
ideal option is DISORT 8-stream with 1 cm$^{-1}$ resolution, there is a trade-off between speed and accuracy.
**Fig. 6** compares DISORT simulated radiances at three band resolutions. We use two spectral ranges of
0.4 – 0.5 µm and 1.5 – 2.0 µm to illustrate the differences. **Fig. 6** shows that the coarser band resolution
has smoothed out the radiance variations. The 15 cm$^{-1}$ has the smoothest curve among the three, and 1
cm$^{-1}$ shows more variations than the other two. Another (scientific) criteria for selecting the spectral



resolution is the ability to resolve/match the relative spectral response function (SRF) of a sensor. For
example, the SRFs of channels 1-6 of ABI are given at every 1 cm$^{-1}$.
Accordingly, we have chosen the 1 cm$^{-1}$ band model for the MODTRAN radiance simulations. Performed
were also radiance simulations from different multiple scattering models at 1 cm$^{-1}$ resolution. The whole
spectrum of 0.2 – 4 μm was separated to 14 sections so that the differences can be assessed clearly. For
wavelength below 0.3 μm and beyond 2.5 no discernible differences were found among Isaacs, DISORT
2-, 4-, and 8-strem, and Scaled Isaac. The largest differences occurred in the spectral range of 0.4 – 1.0
μm. Scaled Isaac 8-stream follows DISORT 8-stream closely across the whole spectral range; the Scaled
Isaac method provided near-DISORT accuracy with the speed of Isaacs. Thus, the MODTRAN4.3
simulations for GOES-R ABI were set-up with Scaled Isaac 8-stream with 1 cm$^{-1}$ band resolution.
For illustration, in **Fig. 7** compared are radiances simulated by Isaac 2 stream, Scaled Isaac, and DISORT-
4 stream for the case of Relative Azimuthal Angle=1.9º, View Angle=76.3º, Solar Zenith Angle=87.2º.
The lines are differences between various settings and DISORT-8 stream (e.g. Isaacs minus DISORT-8).
Isaac has the least accuracy since it is oversimplified, 4-stream showed some improvements when
compared with Isaac while still has large differences for 0.4 μm and is still computationally demanding.
Scaled Isaac provides the smallest differences between DISORT-8. **Fig. 6** (lower) zoomed in to the large
difference area of 0.3-0.35 μm which indicates that Scaled Isaacs still provides satisfactory results.

**2.6 Regression methodologies**

We have derived coefficients of regression using a non-constrained and constrained least-square curve
fitting methods of Matlab "stepwisefit" and "lsqnonneg". The first one does stepwise regression by adding
terms to and removing terms from a multilinear model based on their statistical significance. It may give
negative coefficients that results in a negative TOA flux, which is not a physically valid result.
Subsequently, we have re-derived all the coefficients with "lsqnonneg" which can solve a linear or
nonlinear least-squares (data-fitting) problem and produce non-negative coefficients.





To ensure that information from all channels is used and avoid the complex cross-correlation
problem, it was opted to generate Narrow to Broad (NTB) coefficients for each ABI channel
separately (using "lsqnonneg"). These channel specific NTB coefficients are applied to each channel
to convert ABI narrow-band reflectance to extended band. The final broad-band TOA reflectance is
taken as the weighted sum of all 6-channel specific broad-band reflectance. The logic behind this
approach is the assumption that the narrow-band reflectance from each channel is a good
representative for a limited spectral region centered around the channel and the total spectral
reflectance is dominated by the spectral region that contains the most solar energy.
To generate "separate-channel" NTB coefficients, each narrow-band ABI channel reflectance is
converted to a reflectance $\rho_{bb,i}$ separately,
$$\rho_{bb,i}(\theta_0, \theta, \phi) = c_{0,i}(\theta_0, \theta, \phi) + c_{1,i}(\theta_0, \theta, \phi) * \rho_{nb,i}(\theta_0, \theta, \phi) \qquad (4)$$
where $\rho_{bb,i}$ is the band reflectance for an interval around each channel $i$; $c_{0,i}$ and $c_{1,i}$ are regression
coefficients for channel $i$. These regression coefficients are derived separately for various combination of
surface, cloud and aerosol types; The total shortwave broad band ($0.25 - 4.0\mu m$) reflectance $\rho_{bb}^{est}$ is
obtained by taking the weighted sum of all 6 $\rho_{bb,i}$ reflectance
$$\rho_{bb}^{est}(\theta_0, \theta, \phi) = \sum_i \rho_{bb,i}(\theta_0, \theta, \phi) \frac{S_{0,i}}{S_0} \qquad (5)$$
Here, $S_0$ and $S_{0,i}$ are total solar irradiance and band solar irradiance for each channel, respectively. Band
edges around the six ABI channels are: 49980, 18723, 13185, 9221, 6812, 5292, 2500 cm$^{-1.}$ The
corresponding band solar irradiance values are 364, 360, 287, 168, 91, 87 W m$^{-2}$. **Fig. 8** shows the sensor
response function (SRF) and locations of the six ABI channels.
Coefficients are generated for clear condition and 3 types of cloudy conditions. Comparison between ABI
TOA flux and CERES products are shown in Figure 9. The "separate-channel" coefficients work well for
predominantly clear sky. Differences are somewhat more scattered for cloudy cases. The reason may be
due to the fact that the ABI observation time and CERES product time do not match perfectly since cloud
condition change quickly.






**3.0 Data used**


**3.1 Satellite data for GOES-16 and GOES17**


The GOES Imager data were downloaded from https://www.bou.class.noaa.gov/ and the SRF from
https://ncc.nesdis.noaa.gov/GOESR/ABI.php


* The CODC data were not always available from CLASS and had to be obtained from NOAA/STAR
temporary archives. Also, not all the required angular information needed for implementation of
regressions was available online and had to be recomputed.

**3.2 Reference data from CERES and-FLASHFlux Level2 (FLASH_SSF) Version 3C**

260

Near real-time CERES fluxes and clouds in the SSF format are available within about a week of
observation (Kratz et al., 2014). They do not use the most recent CERES instrument calibration and thus
contains some uncertainty. Before GOES data were transferred to the Comprehensive Large Array-data
Stewardship System (CLASS) system, the NOAA/STAR archive was holding new data for about a week.
Therefore, the initial evaluations had to be done only with data that overlapped in time. The CERES data
known as the FLASHFlux Level2 (FLASH_SSF) were available almost in real time and did overlap with
GOES. These data were downloaded from:
https://ceres.larc.nasa.gov/products.php?product=FLASHFlux-Level2
Due to these limitations the early comparison was done between ABI data as archived at NOAA/STAR
and the FLASHFlux products. The archiving of GOES-R at the NOAA Comprehensive Large Array-data
Stewardship System (CLASS) started only in 2019however, it contains data starting from 2017. Once the
CLASS archive became available, we have augmented GOES-16 cases with observations from GOES-
17; only those cases will be shown in this paper.



274

## 3.3 Data preparation

276

The CERES FLASHFlux_SSF data are re-gridded to match ABI spatial resolution by bi-linear interpolation method from the Earth System Modeling Framework (ESMF) package. The full description of the package can be found via http://earthsystemmodeling.org/regrid/#overview. The time difference between CERES FLASHFlux_SSF and GOES-16 data must be less than ±5 min. e.g., if the GOES-R scanning time is 18:51, then the scripts search the FLASHFLUX points between 18:46~18:56, and use the re-gridding method mentioned above to remap the FLASHFLUX to the GOES-R (2 km) domain. Several cases will be illustrated.

The statistics are based on all available points in overlap area. No outliers are removed. All sky, clear sky only, and cloudy only are compared for dates randomly selected. The hour was selected when both GOES-16 and GOES-17 had overlap with CERES FLASHFlux_SSF (Aqua/Terra) data. The coefficients for GOES-17 were obtained by replacing the GOES-16 spectral response function (SRF) by the GOES-17 SRF. All the regressions have been repeated for GOES-17. The GOES-17 SRF was downloaded from https://ncc.nesdis.noaa.gov/GOESR/ABI.php. Simultaneous evaluation for both satellites was performed. The evaluations against the CERES FLASHFlux_SSF data is at footprint scale and covers one hour. The GOES-16 and 17 CONUS data have 5 min intervals, and there are 12 cases in one hour; this requires to test each case independently to find the best time match with CERES FLASHFlux_SSF.

293

## 4.0 Results

295

## 4.1 Comparison between ABI TOA fluxes to those from CERES and/or FLASHFlux

The FLASHFLUX is in footprint format thus it is a variable in time [flux (time)].

In the matching, points that fall in the ±5 min interval of the GOES-R scanning time are used using bilinear interpolation method to get the values for GOES-R domain (e.g., if the GOES-R scanning time



is 18:51, then the scripts search the FLASHFLUX points between 18:46~18:56, and use bilinear
interpolation method to do the remapping to GOES-R (2 km) domain). A case for 2019/12/26 (doy 360)
UTC 19:36 is illustrated in **Fig. 10.**
The derivation and evaluation of TOA radiative fluxes as simulated for any given instrument are quite
challenging. In principle, there is a need to account for all possible changes in the atmospheric and surface
conditions one may encounter in the future. Yet, to know what these conditions are at the time of actual
observation when there is a need to select the appropriate combination of variables from the simulations,
is a formidable task. Therefore, error can be expected due to discrepancies between the actual conditions
and the selected simulations and these are difficult to estimate. The approach we have selected is based
on high-quality simulations using a proven and accepted radiative transfer code (MODTRAN) of known
configurations and a wide range of atmospheric conditions. We have also selected the best available
estimates of TOA radiative fluxes from independent sources for evaluation. However, the matching
between different satellites in space and time is challenging. In selecting the cases for evaluation, we have
adhered to strict criteria of time and space coincidence as described in section 3.3.
We have conducted several experiments to select an appropriate regression approach to the NTB
transformation ensuring that non-physical results are not encountered. Based on the samples used in this
study the differences found for Terra and GOES-16 were in the range of -0.5-(-12.10) for bias and 43.28-
82.09 for standard deviation; for Terra and GOES-17 they were 10.81-48.17 and 70.25-109.19,
respectively. For Aqua and GOES-16 they were 7.02-29.66 and 45.55-109.08 respectively while for Aqua
and GOES-17 they were 0.19-26 and 53.08-94.90, respectively (all units are W m$^{-2}$). The evaluation
process revealed the challenges in undertaking such comparisons. Both estimates of TOA fluxes (CERES
and GOES) do no account for seasonality in the land use classification; the time matching for the different
satellites is important and limits the number of samples that can be used in the comparison. Based on the
results of this study recommendation for future work include the need to incorporate seasonality in land
use and spectral characteristic of the various surface types. Possible stratification by season in the
regressions could also be explored.



## 5.1 Causes for differences between ABI and CERES TOA fluxes

### 5.1.1 Differences in surface spectral reflectance

In the MODTRAN simulations we use the spectral reflectance information on various surface types as provided by MODTRAN. MODTRAN version 4.3.1 contains a collection of spectral surface reflectance dataset from the Moderate Spectral Atmospheric Radiance and Transmittance (MOSART) model (Cornette et al., 1994) and others from Johns Hopkins University Spectral Library (Baldridge et al., 2009). When doing simulation, we call the built-in surface types and use the provided surface reflectance. As such, the spectral dependence of the surface reflectance used in the simulations and matched to the CERES surface types may not be compatible with the classification of CERES.

### 5.1.2 Issues related to surface classification

Another possible cause for differences between the TOA fluxes is the classification of surface types as originally identified by the IGBP and used in the simulations. No seasonality is incorporated in the surface type classification and the impact can be illustrated in the following case study. Simulation results for surface type 8 (open shrub) have been checked in depth. The average simulated broad-band reflectance is around 0.2. The regression residual for this surface type is reasonably small for sun angle <80 degrees, namely, the fitted broad-band reflectance is very close to the simulated broad-band reflectance. This would indicate that the regressions are performing properly. However, when we applied the regression coefficient to the GOES-16 ABI observations, the calculated TOA broad-band reflectance was around 0.45, which seemed too high. To explain why the coefficient for channel 6 for "open shrub" was high we illustrate the filter function for channel 6 and spectral albedos for open shrub, desert, woody savanna and grassland in **Fig. 14.**



In **Fig. 15** we show the TOA fluxes for the entire domain using the original IGBP classification (open
shrub) in the area of interest and subsequent replacement with a desert surface. Due to seasonal changes
in surface properties, "Desert" classification may be more appropriate for the surface type at the time of
the observations. This would indicate the need for introducing seasonal variability in the classification of
surface types before one selects the representative NTB transformations.

**5.1.3 Issues related to match-up between GOES-R and CERES**

Both Terra and Aqua have sun-synchronous, near-polar circular orbits. Terra is timed to cross the equator
from north to south (descending node) at approximately 10:30 am local time. Aqua is timed to cross the
equator from south to north (ascending node) at approximately 1:30 pm local time. The periods for Terra
and Aqua are 99 and 98 minutes, respectively. Both have 16 orbits per day. CERES on Terra and Aqua
optical FOV at nadir is 16 x 32 or 20 km resolution. Terra passes CONUS during 03-06 UTC (US night
time), 16-20 UTC (US day time), and Aqua passes CONUS during 07-11 UTC (US night time), 18-22
UTC (US day time).
Both Terra and Aqua have an instantaneous FOV values at SWATH level. There is no
perfect overlap, temporally or spatially with ABI data. The ABI radiance and cloud data are on a regular
grid of 2*2 km over CONUS at each hour. To use CERES data for evaluation of ABI, there is a need to
perform collocation in both time and space.

**6.0 Summary**

Critical elements of an inference scheme for TOA radiative flux estimates from satellite observations are:
1) transformation of narrowband quantities into broadband ones;
2) transformation of bi-directional reflectance into albedo by applying Angular Distribution Models
(ADMs). In principle, the order in which these transformations are executed is arbitrary. However, since



well established, observation-based broadband ADMs derived from the Clouds and the Earth's Radiant
Energy System (CERES) project already exist, the logical procedure is to do the NTB transformation on
the radiances first, and then apply the ADM. This is the sequence that has been followed here. While the
road map to accomplish above objectives seems well defined, reaching the final goal of having a stable
up-to-date procedure for deriving TOA radiative fluxes from a new instrument like the ABI on the new
generation of GOES satellites is quite complicated. The process of preparing for the usefulness of a new
satellite sensor needs to be done in advance, the final configuration of the instrument becomes known at
a much later stage. As such, the evaluation of the new algorithms is in a fluid stage for a long time.
Agreement or disagreement with know "ground truth" is not fully informative on the performance of the
new algorithms to estimate desired geophysical parameters. Additional complication is related to the lack
of maturity of basic information needed in the implementation process, such as a reliable cloud screened
product which in itself is in a process of development and modifications. The "ground truth", namely, the
CERES observations are also undergoing adjustments and recalibration. As such, the process of deriving
best possible estimates of TOA radiative fluxes from ABI underwent numerous iterations to reach its
current status. An effort was made to deal the best way possible with the fluid situation. All the evaluations
against CERES were repeated once the ABI data reached stability and were archived in CLASS and we
used the most recent auxiliary information. The prominence of certain issues surfaced from this study
itself. One example is the sensitivity to land classification which currently is static. Another issue is
related to the representation of real time aerosol optical depth which is important under clear sky
conditions. It is believed that only now when NOAA/STAR has a stable aerosol retrieval algorithm, it
would be timely to address the aerosol issue in the estimation of TOA fluxes under clear sky.

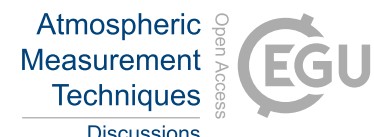

Data availability. The data are available upon request from the corresponding author.
Author contributions. The investigation and conceptualization were carried out by RTP, IL and JD. YM
and WC developed the software. RTP prepared the original draft. All authors contributed to the writing,
editing and review of the publication.
Competing interests. The authors declare that they have no conflict of interest.
Disclaimer. Publisher's note: Copernicus Publications remains neutral with regard to jurisdictional claims
in published maps and institutional affiliations.
Acknowledgements. We acknowledge the benefit from the use of the numerous data sources used in this
study. These include the Clouds and the Earth's Radiant Energy System (CERES) teams, the Fast
Longwave and Shortwave Radiative Flux (FLASHFlux) teams, the
University of Wisconsin-Madison, Space Science and Engineering Center, Cooperative Institute for
Meteorological Satellite Studies (CIMSS) for providing the SeeBor Version 5.0 data
(https://cimss.ssec.wisc.edu/training_data/, and the final versions of the GOES Imager data were
downloaded from https://www.bou.class.noaa.gov/. Several individuals have been involved in the early
stages of the project whose contribution led to the refinements of the methodologies. These include M.
M. Woncsick and Shuyan Liu.

Financial support. This research was supported by NOAA/NESDIS GOES-R Program under grants
5275562 1RPRP_DASR and 275562 RPRP_DASR_20 to the University of Maryland.




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





# Tables

Table 1.        Relevant information for the derivation of SW fluxes from selected satellites:
channel information and spectral bands for ABI.

| ABI Band # | Channel | Spectral band ($\mu m$) |
|---|---|---|
| 1 | VIS 0.47 | 0.45-0.49 |
| 2 | VIS 0.64 | 0.60-0.68 |
| 3 | VIS 0.86 | 0.847-0.882 |
| 4 | NIR 1.38 | 1.366-1.380 |
| 5 | NIR 1.61 | 1.59-1.63 |
| 6 | NIR 2.26 | 2.22-2.27 |









Table 2.    Surface classification description for IGBP 18 types, IGBP 12 types, CERES clear sky 6
483              types, and NTB cloudy sky 4 types

| IGBP (18 types) | IGBP (12 types) | CERES clear-sky (6 types) | NTB cloudy-sky (4 types) |
|---|---|---|---|
| Evergreen Needleleaf | Needleleaf Forest | Mod-High Tree/Shrub | Land |
| Evergreen Broadleaf | Broadleaf Forest | | |
| Deciduous Needleleaf | Needleleaf Forest | | |
| Deciduous Broadleaf | Broadleaf Forest | | |
| Mixed Forest | Mixed Forest | | |
| Closed Shrublands | Closed Shrub | | |
| Open Shrublands | Open Shrub | Dark Desert | |
| Woody Savannas | Woody Savannas | Mod-High Tree/Shrub | |
| Savannas | Savannas | Low-Mod Tree/Shrub | |
| Grasslands | Grasslands | | |
| Permanent Wetlands | | | |
| Croplands | Croplands | | |
| Urban and Built-up | Open Shrub | Dark Desert | Desert |
| Cropland Mosaics | Croplands | Low-Mod Tree/Shrub | Land |
| Snow and Ice | Snow and Ice | Snow and Ice | Snow and Ice |
| Bare Soil and Rocks | Barren and Desert | Bright Desert | Desert |
| Water Bodies | Ocean | Ocean | Water |
| Tundra | Grasslands | Low-Mod Tree/Shrub | Land |








Table 3. The various classes for which NTB coefficients are generated.

| Parameter | Clear condition | Cloudy condition |
|---|---|---|
| Aerosol or cloud type | 6 aerosol types (rural, maritime, urban, tropospheric, fog, desert) | 3 cloud types (cirrus, stratocumulus, altostratus) |
| Optical depth (OD) | Typical VIS (km) values for each aerosol types (no OD grid for each aerosol type). Rural: 23, maritime: 23, urban: 5, tropospheric: 50, fog: 0.2, desert: (default VIS for wind speed 10m/s) | Cirrus: [0, 0.8, 1.2, 1.8, 3.2] Stratocumulus: [0, 0.8, 1.2, 1.8, 3.2, 5.8, 8.2, 15.8, 32.2, 51.8, 124.2] Altostratus: [0, 15.0, 30.0, 50.0, 80.0] |
| Surface type | 12 IGBP surface types | 4 types (Water, Land, Desert, Snow/Ice) |








Table 4. Angles used in simulations. To be consistent with what is presented in the

493       ABI Shortwave Radiation Budget (SRB) Algorithm Theoretical Basis Documents (ATBD) (Laszlo

494       et al, 2018) the additional angles used in the simulations are not given in this Table.

| Angle Type | Angles |
|---|---|
| Solar Zenith Angle [°] | 0.0, 12.9, 30.8, 41.2, 48.3, 56.5, 63.2, 69.5, 75.5, 81.4, 87.2 |
| Satellite Zenith Angle [°] | 0.0, 11.4, 26.1, 40.3, 53.8, 65.9, 76.3 |
| Azimuth Angle [°] | 0.0, 1.9, 10.0, 24.2, 44.0, 68.8, 97.6, 129.3, 162.9, 180 |









499              Table 5. MODTRAN simulation speed test (CPU MHz 2099.929).

| Algorithm | Stream | Band Resolution (cm⁻¹) | Speed (~seconds) |
|---|---|---|---|
| Isaacs | 2 | 1 | 40 |
| DISORT | 2 | 1, 5, 15 | 280, 70, 30 |
| | 4 | 1, 5, 15 | 560, 120, 40 |
| | 8 | 1, 5, 15 | 930, 300, 110 |
| Scaled Isaac | 2 | 1, 5, 15 | 30, 10, 6.67 |
| | 4 | 1, 5, 15 | 30, 10, 6.67 |
| | 8 | 1, 5, 15 | 30, 10, 6.67 |







Table 6.        Details on data used as input for calculations.

| Short Name | Long Name | MODE | ABI-Channel | Scan Sector | Spatial Resolution |
| --- | --- | --- | --- | --- | --- |
| RadC | L1b Radiance | M6 | C01-C06 | CONUS | 5000x3000 |
| AODC | L2 Aerosol | M6 | -- | CONUS | 2500x1500 |
| ACMC | L2 Clear Sky Masks | M6 | -- | CONUS | 2500x1500 |
| ACTPC | L2 Cloud Top Phase | M6 | -- | CONUS | 2500x1500 |
| CODC* | L2 Cloud Optical Depth | M6 | -- | CONUS | 2500x1500 |









Table 7. Statistical summary for all selected cases intercompared at instantaneous time

511        scale.

| Case | CERES | GOES-R | Corr | Bias | Std | RMSE | N |
|------|-------|--------|------|------|-----|------|---|
| 09/13 2019 UTC 20 | Terra | G16 | 0.87 | -12.10 | 82.09 | 82.98 | $0.13 \times 10^6$ |
| | | G17 | 0.71 | 48.17 | 108.19 | 118.42 | $1.73 \times 10^6$ |
| | Aqua | G16 | 0.76 | 17.38 | 109.08 | 110.45 | $1.46 \times 10^6$ |
| | | G17 | 0.73 | 26.00 | 81.96 | 85.98 | $0.53 \times 10^6$ |
| 09/21 2019 UTC 19 | Terra | G16 | 0.85 | 6.78 | 66.66 | 67.00 | $0.35 \times 10^6$ |
| | | G17 | 0.83 | 26.41 | 87.64 | 91.57 | $1.75 \times 10^6$ |
| | Aqua | G16 | 0.82 | 29.66 | 105.09 | 109.20 | $1.67 \times 10^6$ |
| | | G17 | 0.76 | 6.03 | 94.70 | 94.89 | $0.15 \times 10^6$ |
| 09/30 2019 UTC 19 | Terra | G16 | 0.88 | 4.49 | 64.79 | 64.94 | $0.40 \times 10^6$ |
| | | G17 | 0.80 | 19.35 | 86.41 | 88.55 | $1.74 \times 10^6$ |
| | Aqua | G16 | 0.81 | 19.99 | 99.98 | 101.96 | $1.67 \times 10^6$ |
| | | G17 | 0.70 | 1.22 | 94.90 | 94.91 | $0.12 \times 10^6$ |
| 10/23 2019 | Terra | G16 | 0.86 | 5.84 | 51.44 | 51.77 | $0.35 \times 10^6$ |
| | | G17 | 0.87 | 22.47 | 70.25 | 73.76 | $1.75 \times 10^6$ |
| | Aqua | G16 | 0.89 | 17.10 | 75.95 | 77.85 | $1.67 \times 10^6$ |





| | | | | | | |
|---|---|---|---|---|---|---|
| UTC 19 | | G17 | 0.78 | 8.98 | 72.52 | 73.07 | $0.15 \times 10^6$ |
| 11/08 2019 UTC 19 | Terra | G16 | 0.87 | -0.5 | 43.28 | 43.28 | $0.35 \times 10^6$ |
| | | G17 | 0.82 | 17.18 | 71.27 | 73.31 | $1.75 \times 10^6$ |
| | Aqua | G16 | 0.90 | 10.08 | 71.27 | 71.98 | $1.67 \times 10^6$ |
| | | G17 | 0.68 | 1.53 | 47.55 | 47.58 | $0.15 \times 10^6$ |
| 11/24 2019 UTC 19 | Terra | G16 | 0.79 | 7.98 | 49.10 | 49.75 | $0.35 \times 10^6$ |
| | | G17 | 0.87 | 14.10 | 78.35 | 79.61 | $1.76 \times 10^6$ |
| | Aqua | G16 | 0.82 | 7.63 | 58.68 | 59.17 | $1.67 \times 10^6$ |
| | | G17 | 0.65 | 0.19 | 63.14 | 63.14 | $0.15 \times 10^6$ |
| 12/26 2019 UTC 19 | Terra | G16 | 0.89 | 7.6 | 52.79 | 53.33 | $0.35 \times 10^6$ |
| | | G17 | 0.77 | 10.81 | 73.14 | 73.93 | $1.76 \times 10^6$ |
| | Aqua | G16 | 0.83 | 7.02 | 59.16 | 59.58 | $1.67 \times 10^6$ |
| | | G17 | 0.73 | -1.09 | 53.08 | 53.09 | $0.15 \times 10^6$ |








# Figures

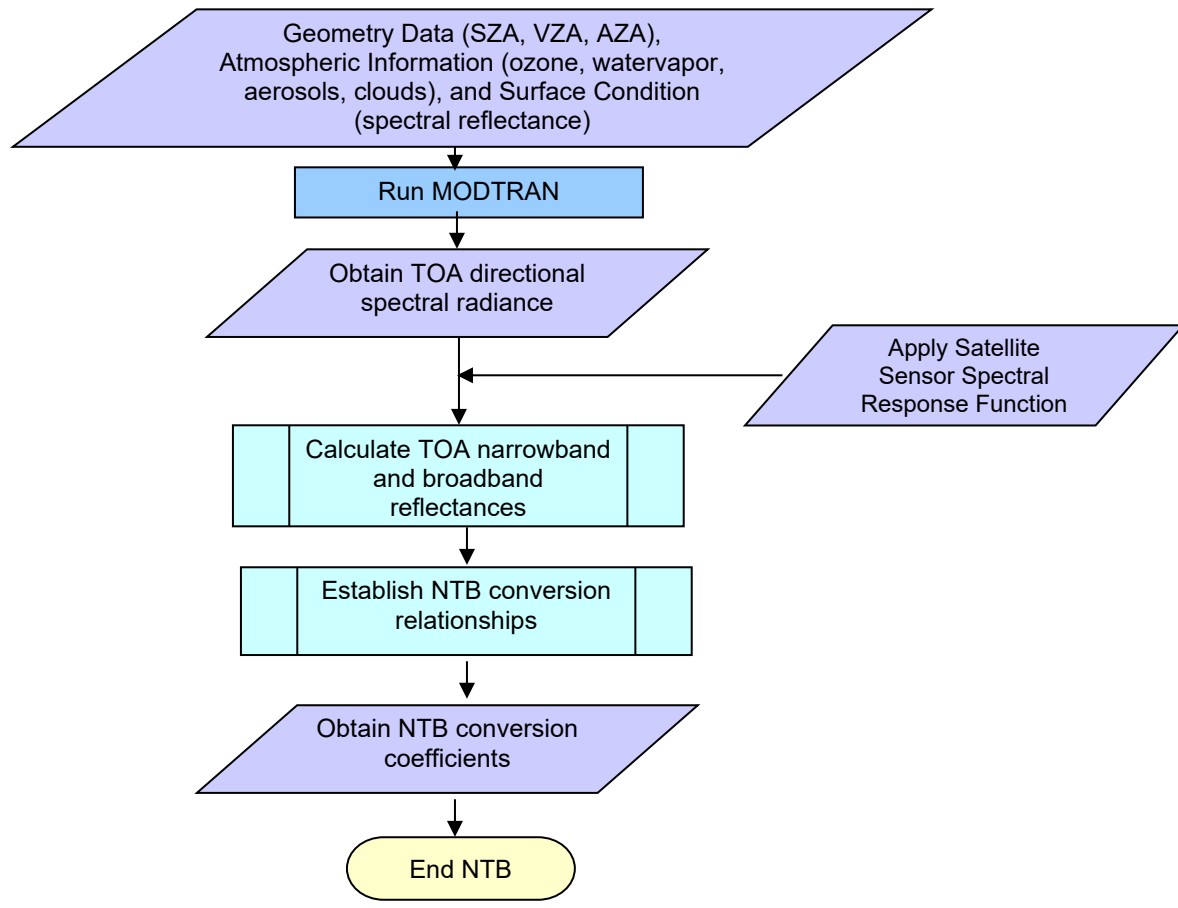

Figure 1. Flowchart of the NTB transformations illustrating the main processing sections.



```
                    ┌─────────────────┐
                    │   Start ADMs    │
                    └────────┬────────┘
                             │
          ┌──────────────────────────────────────┐
          │  Obtain the simulated ADMs based      │
          │  on IGBP surface classifications      │
          └──────────────────┬───────────────────┘
                             │
                Y    ◇ Clear Sky? ◇     N
          ┌───────────┘              └───────────┐
          │                                      │
┌─────────────────────────┐      ┌─────────────────────────────┐
│ Combine the corresponding│      │ Based on cloud phase         │
│ CERES and simulated ADMs │      │ (water, ice) and cloud       │
│ based on IGBP surface    │      │ optical depth intervals      │
│ classifications          │      └──────────────┬──────────────┘
└────────────┬─────────────┘                     │
             │                      ┌─────────────────────────────┐
┌─────────────────────────┐        │ Utilized the CERES cloud    │
│ Obtain the synthesized   │        │ ADMs for surface type of    │
│ Clear-sky ADMs based on  │        │ ocean, low-mod shrub/tree,  │
│ IGBP surface             │        │ mod-high shrub/tree,        │
│ classifications          │        │ desert (bright, dark), and  │
└────────────┬─────────────┘        │ snow/ice                    │
             │                      └──────────────┬──────────────┘
             └──────────────┬──────────────────────┘
                    ┌─────────────────────────┐
                    │ Select the corresponding │
                    │ ADMs based on surface    │
                    │ scene and cloud state    │
                    └────────────┬─────────────┘
                    ┌─────────────────────────┐
                    │ Apply the corresponding  │
                    │ synthesized ADMs to      │
                    │ obtain TOA broadband     │
                    │ albedos                  │
                    └────────────┬─────────────┘
                        ┌──────────────────┐
                        │    End ADMs      │
                        └──────────────────┘
```


523 Figure 2. Schematic illustration of the logic employed to synthesize modeled and observed ADMs.









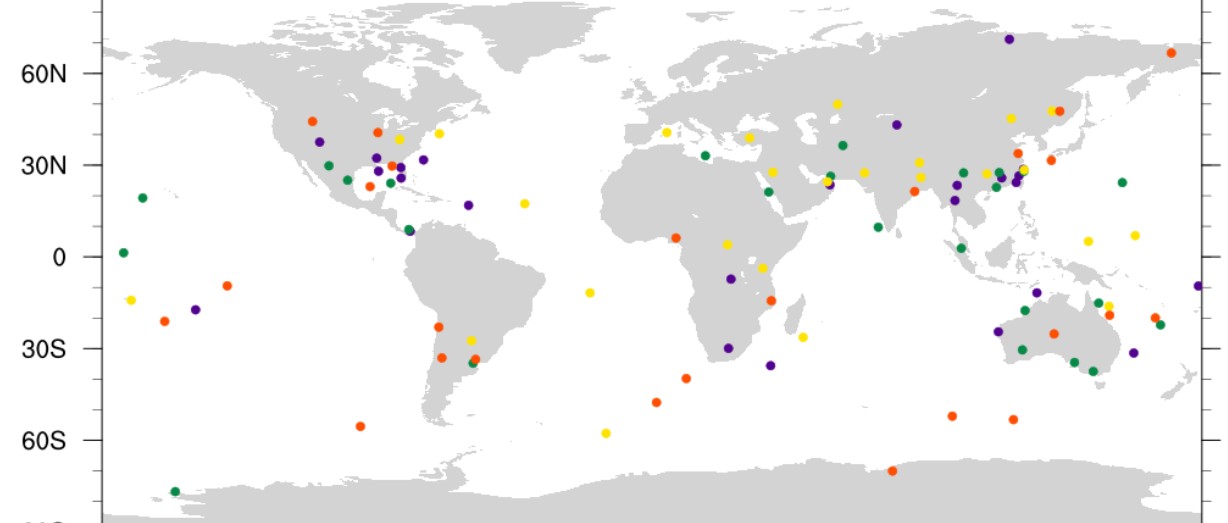


Figure 3. The location of the 100 selected clear sky profiles from SeeBor used in





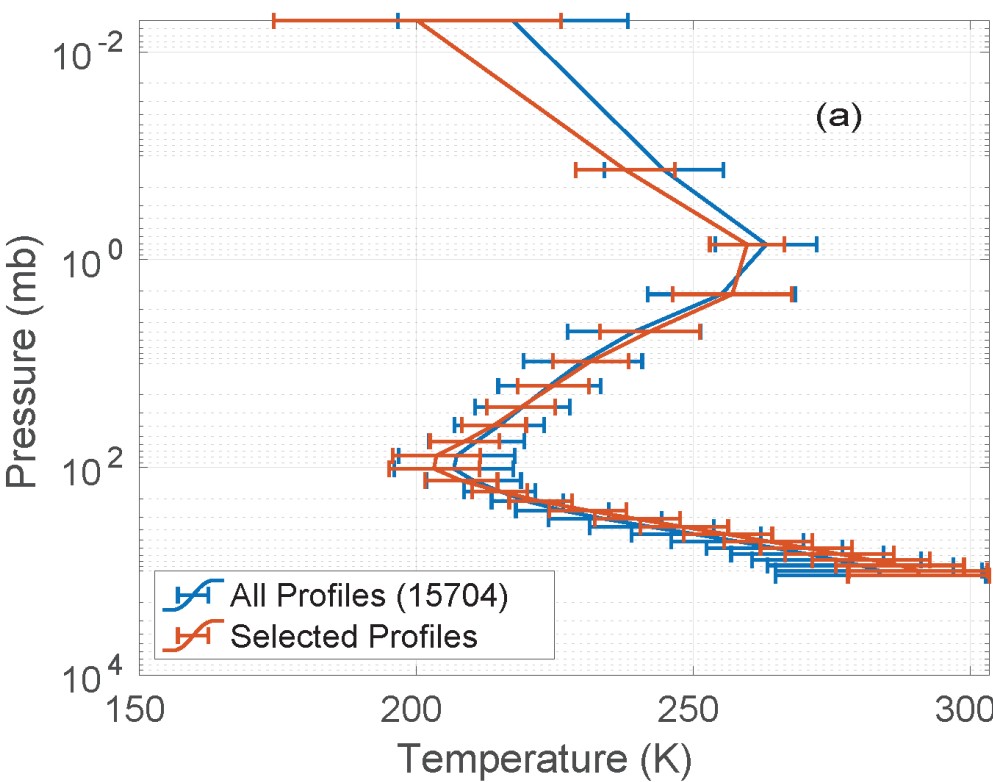






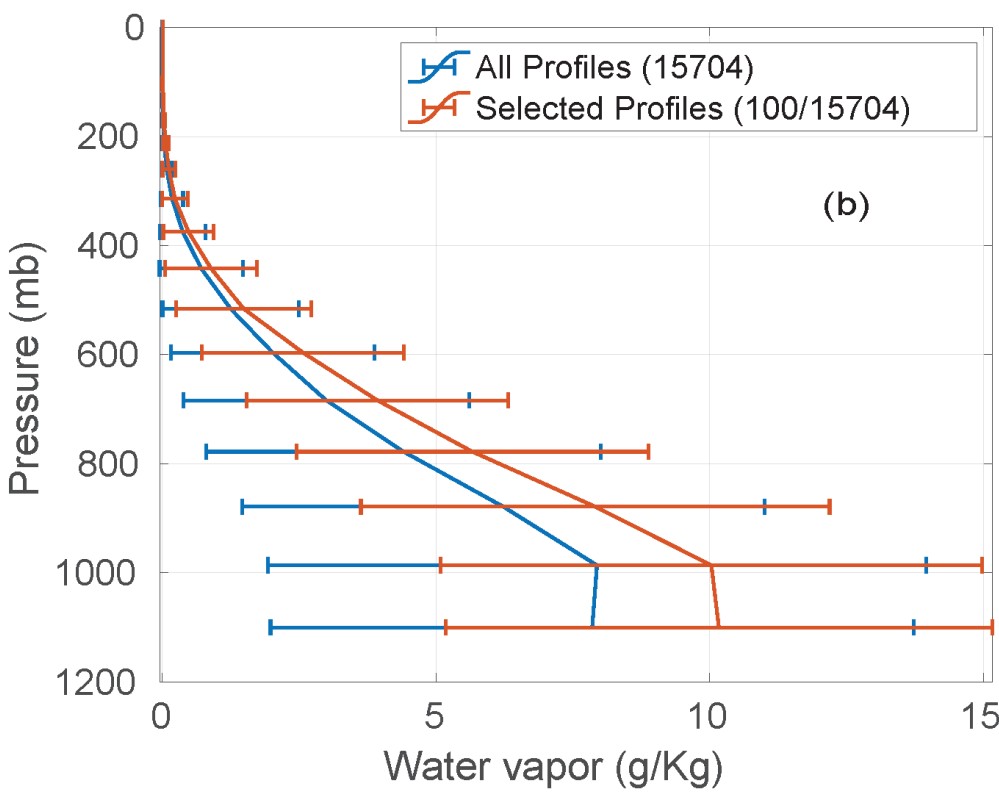






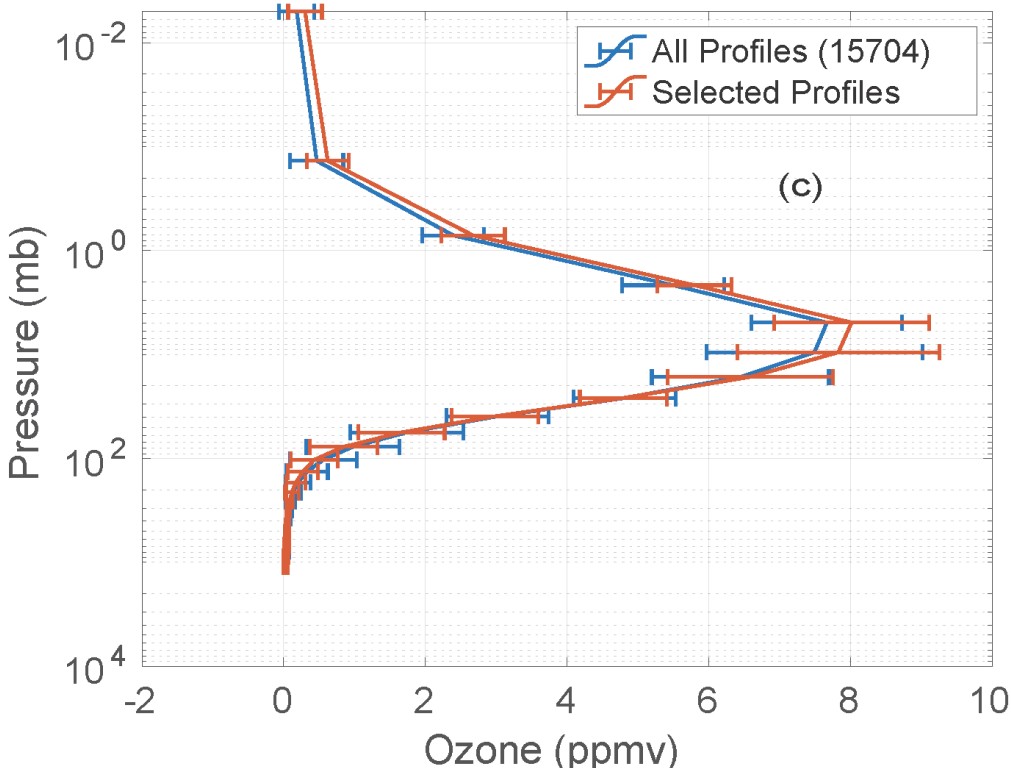



Figure 4.       Profile statistics of: (a) temperature; (b): water vapor; (c) ozone the entire available

sample and the reduced sample used in this study. Error bar is 1 standard deviation (logarithmic scale).







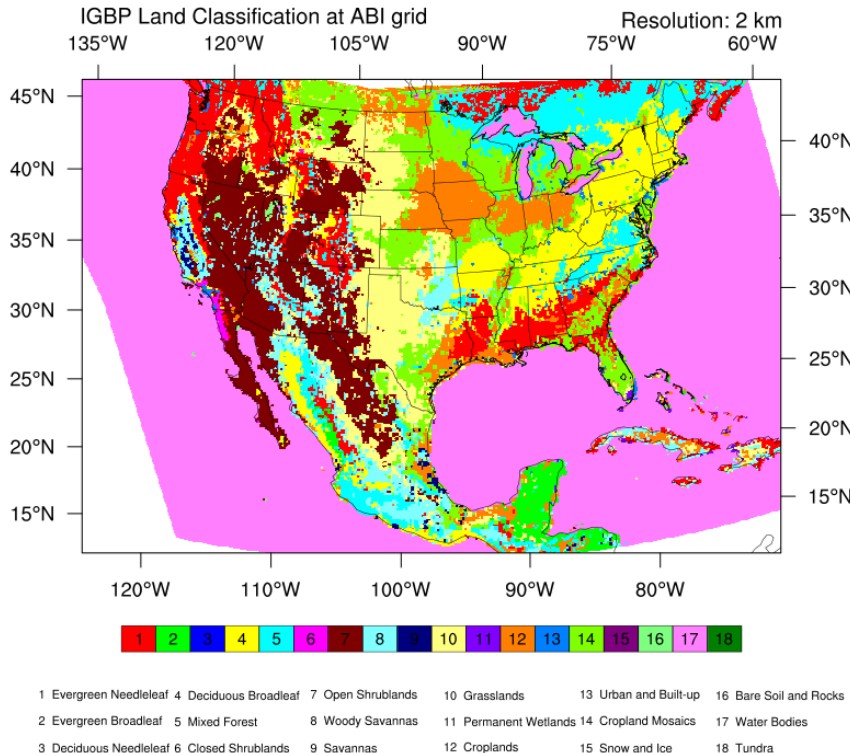


Figure 5. Re-mapped IGBP surface classifications over the CONUS at 2-km ABI grid.









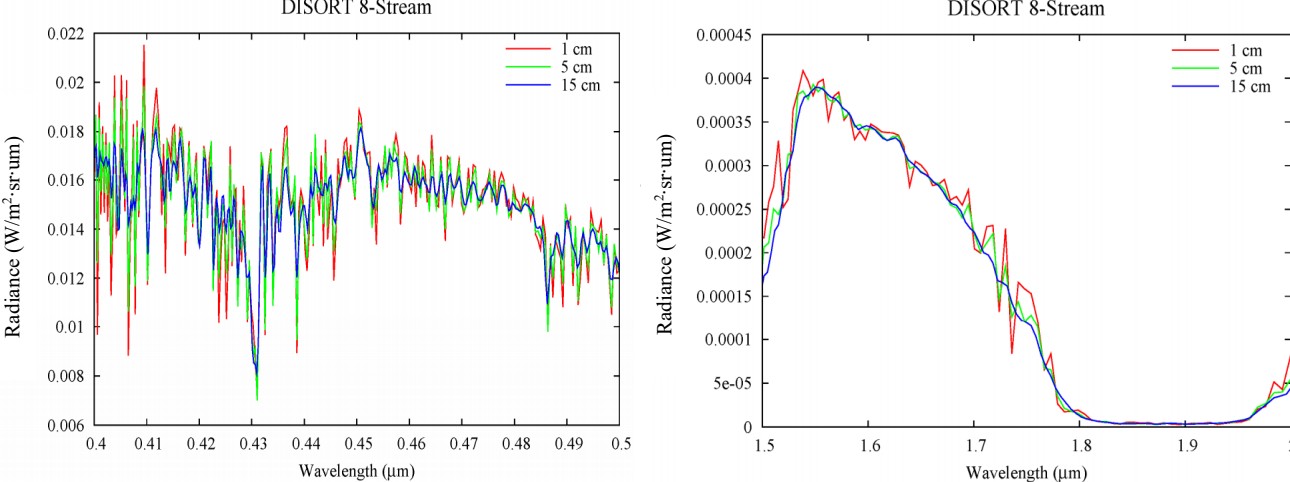


Figure 6. Simulated Radiances from DISORT 8-stream (with 1, 5, and 15 cm[-1] resolution band

model for spectral range of 0.4 – 0.5 µm (left) and 1.5 – 2.0 µm (right).



Figure 7. Radiance differences between various multi-scattering algorithms and DISORT-8 stream. *Upper*: the whole simulated spectrum of 0.2-4 μm; *Lower:* zoom on 0.3-0.35 μm (Relative Azimuthal Angle=1.9º, View Angle=76.3º, Solar Zenith Angle=87.2º).







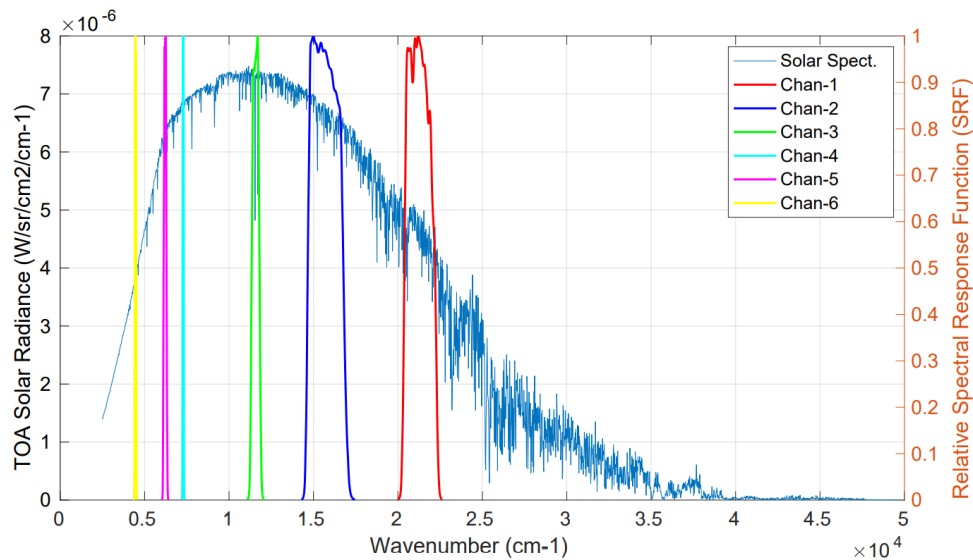


**Figure 8**. Locations of the six ABI channel SRFs. X-axis is wavenumber. Y-axis is solar irradiance.






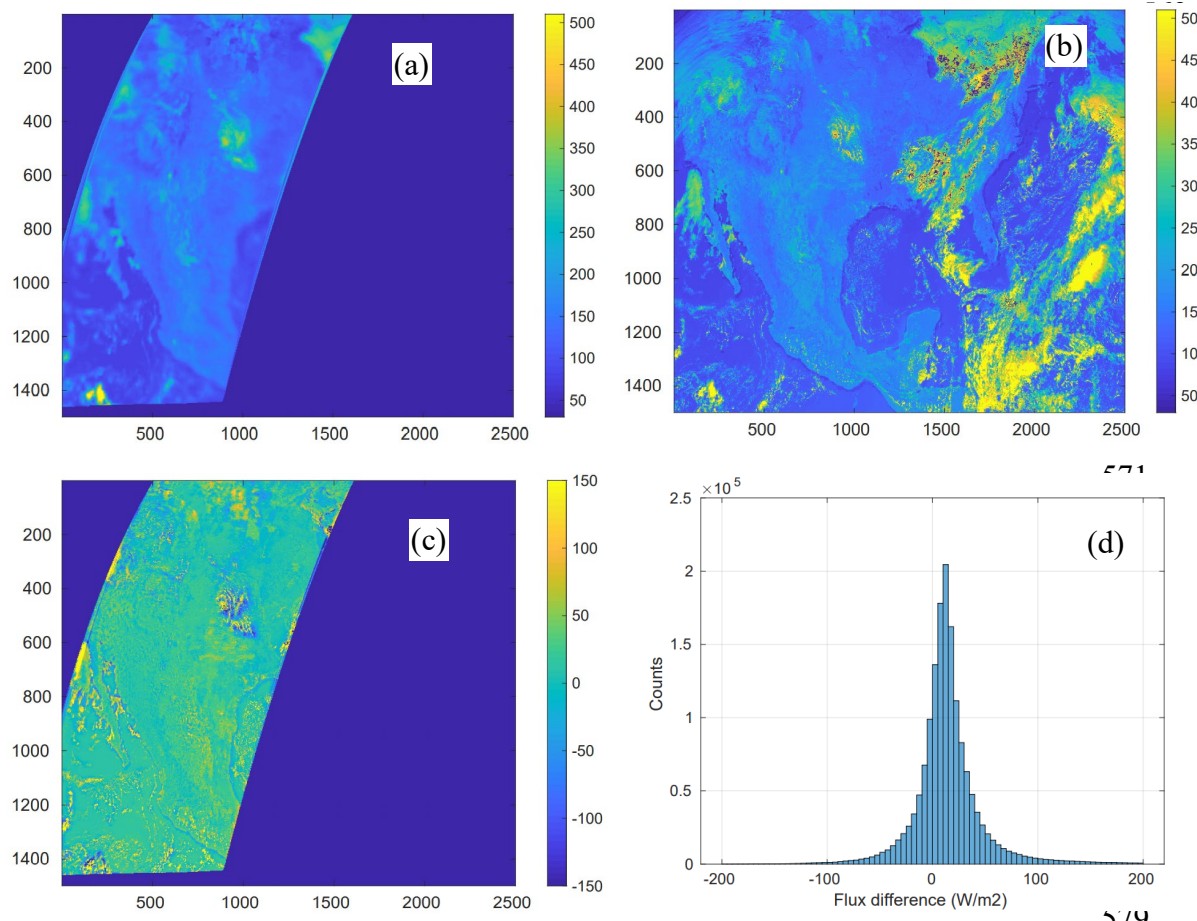

**Figure 9**. Comparison of TOA flux from ABI and CERES based FLASHFlux for 2017/11/25, 17:57Z.
(a) CERES Terra product; (b): results with "separate-channel" coefficients. (c): difference (ABI-
CERES); (d): histogram of ABI-CERES differences.

583

584



585

Figure 10. All sky TOA SW from CERES FLASHFlux/Aqua (a), CERES FLASHFlux/Terra (b), re-
gridded CERES FLASHFlux/Aqua (c), CERES FLASHFlux/Terra GOES-16 (d) and GOES-17 (f) on
12/26/2019 at UTC 19:36.




Figure 11. Frequency distribution of all-sky TOA SW differences between ABI on GOES-16 and CERES

(*Left*) and ABI on GOES-17 and CERES (*Right*) using Aqua (Upper) and Terra (Lower). All observations

were used (clear and cloudy) on 12/26/2019 at UTC 19:36.







Figure 12. Same as Figure 11 but for clear TOA SW differences.






Figure 13. Same as Figure 11 but for cloudy TOA SW differences.









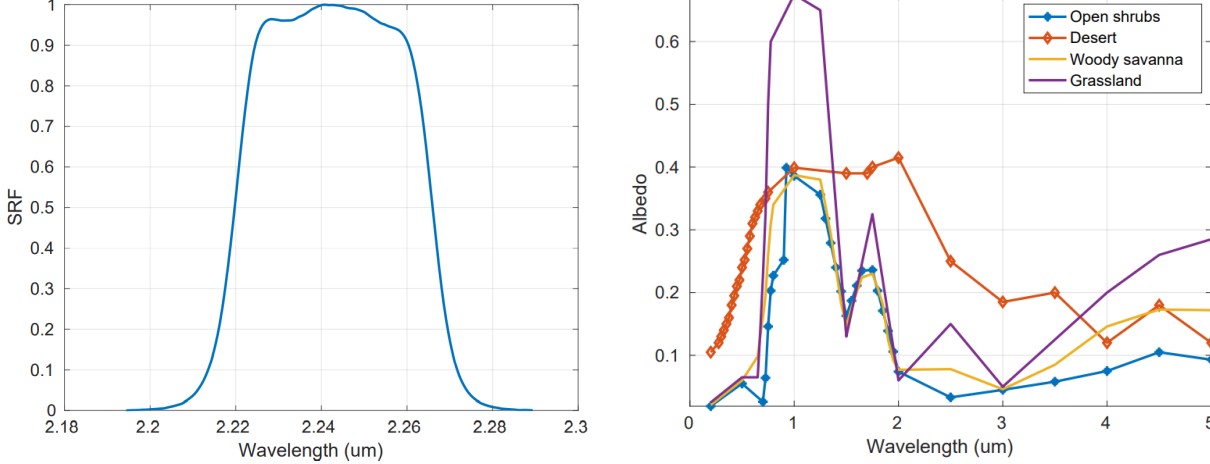


**Figure 14**. *Left:* Sensor response function for ABI channel 6; *Right:* Spectral albedo for desert and open
shrubs. Desert albedo value is much higher than open shrubs at 2.2 μm.



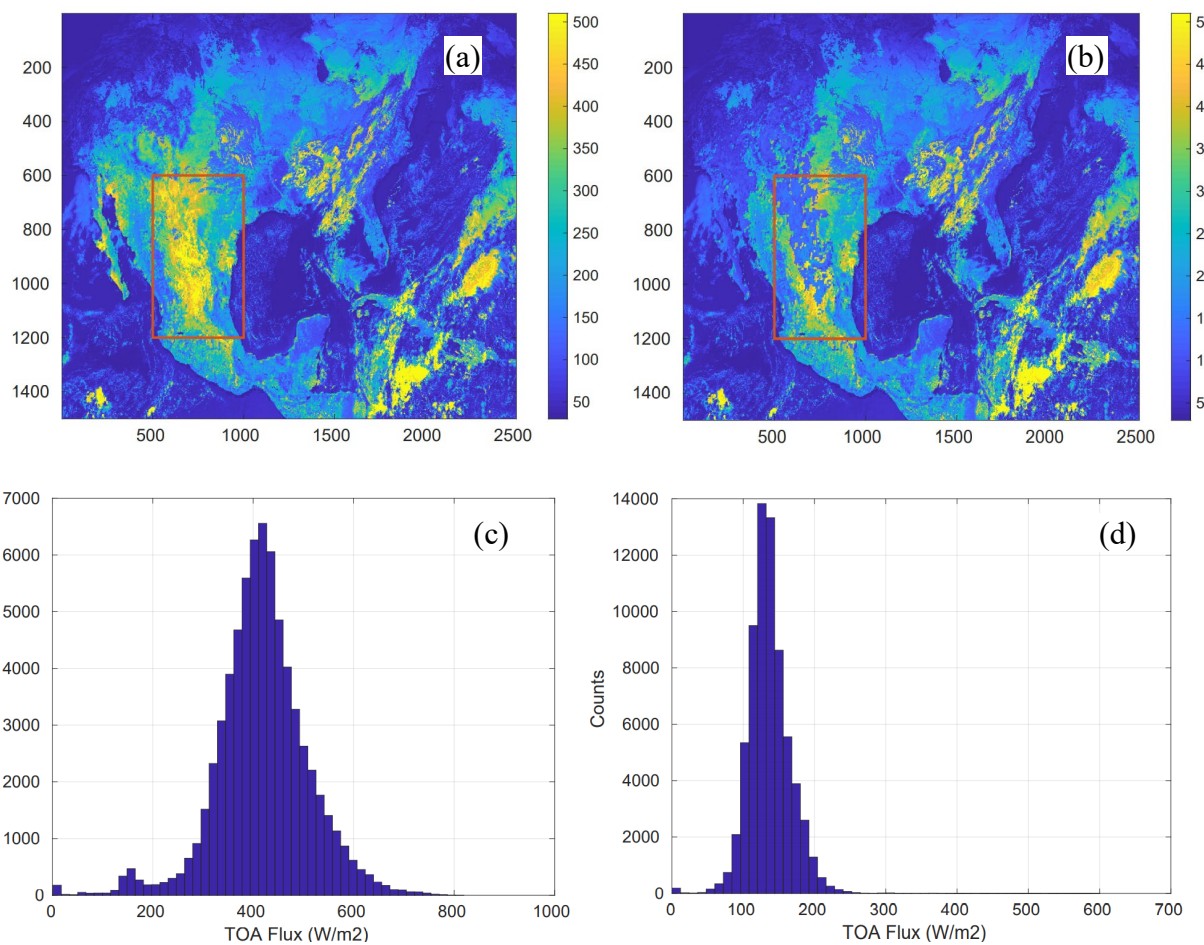

**Figure 15**.    TOA fluxes using two different NTB coefficients: *Left*: used "open shrub" coefficients;
*Right*: "Desert" coefficients. Lower panels show the frequency distribution of TOA fluxes for a reduced
domain (over Mexico in the orange boxes) that includes the open shrub/desert classification. Case time
stamp is 2017/11/25 17:32Z.

