# Peer review of "Top of the Atmosphere Reflected Shortwave Radiative Fluxes from GOES-R"

_Atmospheric Measurement Techniques, 2021_

## Referee Comment (RC1)

**Review of "Top of the Atmosphere Reflected Shortwave Radiative Fluxes from GOES-R" by Pinker et al.**

**21 October 2021**

**Overview**

The manuscript prepared by Pinker et al describes the conversion of radiances from the ABI instrument on GOES-R to SW radiative fluxes. First, a spectral regression is applied to convert narrow band radiances to broadband radiances. Second, angular distribution models are applied to convert the broadband radiances to radiative fluxes. The derived radiative fluxes are compared to those from the CERES FLASHFlux product. Possible reasons for discrepancies are discussed.

This work addresses an important and interesting topic, and I believe that SW radiative fluxes from GOES have the potential to be of great value to the scientific community. However, I have several major concerns as outlined below. In summary, there are significant gaps in the description of the methods that need addressing, and the reasons for differences with CERES data would benefit from some additional analysis. After addressing these concerns, I believe the work would be a good fit for publication in *Atmospheric Measurement Techniques*.

**Major comments**

L99: In order to apply equation 3, there is an assumption that ADMs from observations and simulations for a given scene type belong to the same population. I am not convinced this is the case. If the CERES anisotropic factors and the simulated anisotropic factors are substantially different (eg. due to neglected processes in the simulations such as 3D radiative effects), the weighted average anisotropic factor from equation 3 might end up somewhere in the middle, not representing either. I suggest discussing this caveat, or addressing this issue with a figure showing that the underlying radiances for a challenging scene type largely overlap between the simulations and CERES.

L103: How is it possible to know "m", ie. the number of CERES observations associated with the anisotropic factor for each angular bin? If I understand correctly, the authors are using the existing CERES ADMs derived from the CERES instrument on TRMM (Loeb et al., 2003), combined with their simulations. These CERES ADMs provide anisotropic factors but, to my knowledge, they do not provide the number of observations that were used to derive the anisotropic factors in each angular bin.

L109: What is the "tool" that was developed to select 100 profiles from the original database of 15704? How does it ensure a variety of conditions are represented? Details are needed, otherwise there is no way that the results can be reproduced.

L164-171: Some key information is missing relating to how clouds are included in the simulations. The following should be included in Table 3:

- What are the cloud altitude/pressure boundaries for the 3 cloud types considered?

- What is the phase of each cloud type? I assume cirrus is ice, stratocumulus is liquid. Altostratus is a mixture? What ice optical properties are used in the simulations?

- Are the 3 cloud types always simulated in isolation, or does the set of simulations include combinations ie. multi-layer cloud?

- Is there any attempt to consider cloud fraction?

L264-267: The differences shown in Fig 9c and 9d occur after applying a NTB conversion and then ADMs. The authors claim that the reason for the differences could be the temporal offset between CERES and GOES. I am not convinced. The observations are co-located to within 5min. Not many cloud regimes are drastically changing within 5min at the CERES footprint scale. I expect the uncertainty due to the NTB conversion and ADMs is much larger. For the NTB conversion in cloudy scenes, one possible reason is that the ABI bands do not provide sufficient spectral coverage. Figure 1b in Gristey et al., JClim, 2019 (https://doi.org/10.1175/JCLI-D-18-0815.1) shows SW spectral reflectance variations for different cloud types. Comparing with the ABI bands, I suspect some spectral variations associated with cloud variability are missed. For ADMs in cloudy scenes, the cloud properties must be retrieved for the selection of the correct ADM. Misclassification of cloud properties will therefore result in flux differences. Even if the correct scene type is selected, ADMs have an uncertainty due to within-scene variability and within-angular bin variability. I suggest including discussion of these possible reasons.

L271: This section does not mention the time range/case studies of observations used from GOES-16 and GEOS-17. Some cases are listed in Table 7, but this table is not referenced anywhere in the text. It is not clear if these cases studies encompass all of the data used in the study. Again, this is essential information for anyone interested to reproduce the results.

L293-294: There seems to be an inconsistency here. The previous paragraph states that FLASHFlux was used because the GOES data was only available for about a week, and FLASHFlux is available within that timeframe. Fair enough. But then it is stated that GOES data is now available in the CLASS archive going back to 2017. So, there is no longer a valid reason to perform comparisons against the (less accurate) FLASHFlux data. Is there any reason that the authors cannot perform their analysis using the GOES data from the CLASS archive against the primary CERES L2 SSF product? Maybe I am missing something.

L296: A major step missing from the paper is how the scene properties are determined for the ABI observations. I expected to see details in this section. My understanding is that both the regression coefficients for the NTB conversion and the ADMs are a function of scene type. I see that a fixed surface type is assumed but how are the changing atmospheric properties accounted for when converting the ABI narrow band radiances to broadband fluxes?

L298: I find it strange that the authors decided to perform their comparisons at the ABI spatial resolution by applying a bi-linear interpolation to the CERES data. It would make more sense to aggregate the ABI data and perform comparisons at the CERES footprint scale. By performing comparisons at the coarser of the two scales, non-linearity due to interpolation is not an issue.

L317: There is no reference to Fig 11, 12 or 13 in the text. These figures are key to the findings of the study and should be referred to throughout the results section.

L358-365: I do not necessarily disagree with these comments on possible differences in the surface spectral reflectance, but they are purely speculative and insubstantial. Can any supporting analysis be added? For example, MODIS provides a surface spectral reflectance co-located with CERES on both Aqua and Terra, albeit at a coarse spectral resolution. The observed MODIS surface reflectance could be compared with MODTRAN values, even just for a handful of case studies, to quantify any differences.

L390-400: Again, the text here relating to the temporal offset between GOES and CERES is speculative and would be much better served by some supporting analysis. I suggest including a scatter plot using the same data in Fig. 10. The x-axis would be the temporal offset (ranging from 0 to 5 min) and the y-axis would be the difference between GEOS and CERES. Data points could be colored by scene type. If the temporal offset is an important issue, expect to see a clear positive gradient.

**Minor comments**

L38-44: The first paragraph of the introduction does not really serve a purpose. It is irrelevant for the analysis and does not add much to the manuscript in my opinion. It could be removed.

L51-52: There is a recent review paper on shortwave ADMs that could be cited here: Gristey et al., 2021, https://doi.org/10.3390/rs13132640.

L81: Down arrow in the text is out of place and should be removed.

L129: Are the "surface variables" also part of the SeeBor dataset, or added by the authors? Please clarify when the dataset is first introduced.

L130: Is the surface albedo a single broadband value? If so, how is this combined with the spectral surface albedo used in MODTRAN (discussed later).

L132: There is a positive bias in what variable? At what altitude? Please be more specific. Fig. 4 shows 3 variables. The sign of the temperature bias depends on altitude; the water vapor bias is positive only at lower altitudes; the ozone bias is positive only at higher altitudes.

L135: This section does not mention that the surface type is fixed in time. Implications are discussed later, but it should be stated clearly here since this is where the dataset is first introduced and it is an important aspect of the work.

L146: Under clear-sky, scattering by aerosol is important, but probably not multiple scattering. Most aerosol loadings are dominated by single scattering. Suggest removing "multiple".

L146: In addition to aerosol scattering, what about the role of absorption? The 6 aerosol types considered presumably have different single scatter albedo.

L157: Please provide an explanation of where the number 288,000 comes from. I calculated 6 aerosol types x 12 surface types x 100 profiles = 7200 simulations for clear-sky.

L162: How are the variations at 4 different wind speeds accounted for. The 100 profiles do not include wind speed information. I also assume this is surface wind speed but please clarify.

L176: How is the number of Gaussian quadrature points determined? A sentence or two explaining the use of Gaussian quadrature would help the reader here.

L182: "azimuth angle" should be "relative azimuth angle", I think.

L184: "ignoring spherical geometry" – what does this mean?

L226: 8-stream is used as the baseline/truth in Fig 7, but I do not see any evidence that 8-stream is itself sufficient. If the number of streams was further increased to eg. 16 or 32, would there be any benefit?

L231: Yes, the results for Scaled Isaacs are better than Isaacs, but how to quantify that they are "satisfactory"? I noticed that they are typically much worse than 4-stream DISORT in Fig 7b.

L260: Switching between wavelength and wavenumber is confusing for the reader. Since SW radiation is usually expressed in wavelength, and most of the plots in this study are in wavelength, I strongly suggest converting any instances of wavenumber throughout this manuscript to wavelength for consistency.

L264: "Figure 9" -> "**Fig. 9**" for consistency.

L292: "2019however" – needs fixing.

L301: "must be less than ±5 min". Is this threshold based on any analysis? What is special about 5 min?

L312: At the footprint scale of which instrument, CERES or ABI?

L321-324: This text is a repeat of the previous section and is not needed again here.

L354: Where is the section 5 heading? It jumps straight from section 4.1 to section 5.1. Are there other subsections from section 4 that are missing?

L377: "the calculated broad-band reflectance was around 0.45" – was this for cloud free scenes only?

L379: Agreed that the filter function for channel 6 (Fig 14) could be problematic. But what impact does this have on the total NTB conversion? What is the weight associated with channel 6?

Table 1: The first part of the caption is not necessary.

Table 1: ABI band 3 is NIR, not VIS.

Table 1, column 2: "Central wavelength" would be better than "Channel". Need to include units.

Table 1, column 3: Are these spectral band widths associated with a threshold percent drop off in response?

Table 2: Could be more reader friendly. I suggest ordering the first column so that the groups in the second column are next to each other.

Table 4: "Azimuth angle" -> "Relative azimuth angle".

Table 6: Not referenced anywhere in the text. I do not think it serves a purpose. Suggest removing it.

Table 7: Not referenced anywhere in the text. List of dates and statistics are useful. I suggest keeping the table but making reference to it in the data/results sections.

Fig 1, box 2: "watervapor" -> "water vapor".

Fig 1: Remove arrow leaving bottom box.

Fig 2: Remove floating arrow leaving the left of the first box.

Fig 3: End of caption is missing.

Fig 3: Top of figure seems to be cut off.

Fig 4: Suggest removing "(logarithmic scale)" from the caption. The error bars are plotted on the same (linear) scale.

Fig 6: Are these nadir radiances at TOA? What is the scene type? Need to include this information in the caption.

Fig 7: Wavelength is increasing from right to left, opposite to the previous figure. For consistency, I suggest reproducing this figure with the wavelength increasing from left to right.

Fig. 8: This figure is in wavenumber but others are in wavelength. For consistency, I suggest reproducing this figure in wavelength. Wavenumber could always be included as a second axis along the top of the plot.

Fig 10: Labels need correcting in the caption. (e) is missing, (d) is in the wrong place.

---

## Referee Comment (RC2)

**Summary**:

I believe it is important to publish satellite retrieval product papers to document the algorithms and to properly reference and cite the product. This paper documents the NOAA STAR GOES ABI TOA SW flux product algorithm. The paper highlights the challenges that need to be overcome to develop the GOES-ABI flux product, especially the spectral information needed to compute the broadband TOA albedo from ABI spectral channels. The paper is based on the product ATBD. The paper describes the current algorithm in depth, but the extent of the validation was lacking given that there are 3-years of validation opportunities. Once the product has been fully developed it will be a great asset to the remote sensing community in providing ABI 2-km pixel resolution TOA SW fluxes over the CONUS region. There is much work left to do for a viable product and documenting the progress is worthy of publishing in this journal. I will consider the paper for publication after the following concerns are addressed.

**General comments.**

The authors should be using the official CERES SSF L2 product not the CERES FLASHFlux L2 product for validation. The FLASHFlux product was designed for real-time processing, where many SSF inputs were replaced with real-time datasets, for example the GEOS 5.4.1 reanalysis rdataset was replace by the realtime FPIT dataset. The more realtime dataset algorithm datasets are often revised due to changing input quality. The CERES input datasets were designed for consistency across the record by limiting algorithm changes to avoid discontinuities in the parameter values. Also, the FLASHFlux fluxes do not employ the most up to date CERES instrument calibration coefficients.

I am assuming that the NOAA STAR GOES ABI TOA SW flux product is not available to the public.

I am having a hard time understanding why there is very little validation being performed. Table 7 simply is not sufficient, not even a full year of data is analyzed. It seems that exact time matching is necessary for validation. The ABI scans are every 15/10 minutes providing closely matched ABI and CERES fluxes.

There is no high-resolution TOA SW flux dataset ground truth dataset, agreed, that is the motivation for this product. The CERES dataset provides observed instantaneous SW fluxes at the 20-km nominal resolution. Linear interpolating the CERES footprint center fluxes across the ABI pixels does not represent a valid 2-km flux field. Cloud edges are not distinct.
This implies that the ABI high-resolution TOA SW fluxes should be mapped into the CERES footprint for validation. Or into lower resolution latitude and longitude grid such as performed by Akkermans, T.; Clerbaux, N. Narrowband-to-Broadband Conversions for Top-of-Atmosphere Reflectance from the Advanced Very High Resolution Radiometer (AVHRR). Remote Sens. 2020, 12, 305. https://doi.org/10.3390/rs12020305
Note they also stratify the validation results by IGBP type. The important part of the validation is determining whether the algorithm is not adding an overall bias to the TOA SW fluxes, while trying to reduce the RMS error. The instantaneous RMS error is a function of spatial scale. They also validate a several year's worth of TOA fluxes

The bin/channel regression rely on RTM results that have varying PW and ozone concentration. The PW water above the cloud or clear-sky surface is necessary to predict the NIR water vapor absorption, since none of the ABI band used are located inside absorption bands. It is not clear to me how current algorithm accounts for NIR water vapor absorption? This was unclear in section 2.

I believe the greatest uncertainty in the NTB algorithm is accounting for spectral information. Could the MODIS 2-week surface band BRDF be used in MODTRAN to update the predefined MODTRAN BRDFs? The MODIS BRDF product could be used to account for regional and seasonal variability.

**Specific Comments:**

Line 38 this paragraph seems out of place. Unless this study was used to for ABI channel selection it does not seem relevant.

Line 45. It would be beneficial for the reader to briefly outline the whole algorithm. To discuss both the indirect path and I am assuming a direct path. Perhaps to provide how this algorithm was developed and if it is used in any historical products.

Line 64 Does ground refer to truth dataset or to actual ground observations, since in the summary mentions "ground truth"

Line 95 Are the Kato and Loeb snow ADMs used as part of the CERES Ed2 ADMs?
Kato, S., and N. G. Loeb (2005), Top-of-atmosphere shortwave broadband observed radiance and estimated irradiance over polar regions from Clouds and the Earth's Radiant Energy System (CERES) instruments on Terra, J. Geophys. Res., 110, D07202, doi:10.1029/2004JD005308

Line 97 The Niu and Pinker are theoretical simulations, how do they translate to observation numbers in Eq. 3?

Line 131. I do not see how the Fig. 4 comparison adds value to the paper. The profiles were selected to get a sampling of the diversity of atmospheric profiles found on Earth.

Line 237 Is the Matlab stepwise fit used in the algorithm? If not this should sentence should be left out because it adds confusion.

Fig. 8 Could the spectral boundaries or band edges for each ABI band also be shown in Fig. 8. This way the reader can see the spectral range radiance that is predicted based on a single ABI band.

Line 260 could the band edges be given in μm in the text also.

Section 2.6 Which channel takes into account the bulk of the NIR water vapor absorption?

Line 264 Figure 9 is spelled out, whereas Fig. 8 is not on line 261

Line 266. I would agree that along the cloud edges there would be large differences between ABI and CERES TOA fluxes. These large differences would occur even if there were a perfect algorithm. However, over large spatial domains the ABI and CERES fluxes should be similar.

Table 6 and 7 are not referenced in the text.

Line 267 It would be nice to have statistics for Figure 9 similar to what is in Table 7. I do not see 2017/11/25, 17:57Z Fig 9 statistics with the 2019 statistics in Table 7.

Line 276. This is where Table 6 should be referenced to identify the CODC product

Line 283. The authors should use the CERES SSF Level 2 data, rather than CERES FlashFLUX footprint fluxes. As mentioned in the text, that FlashFLUX does not use the most up to date CERES instrument calibration coefficients. The CERES SSF product is available within 3-months of real-time.

What is limiting the number of validation match ups? Is the issue that your computing resources have limited computer storage that downloading all of the required datasets for ABI pixel level fluxes and comparisons with CERES is not possible after real-time when these products are no longer available at CLASS?

Line 304. The CERES footprint data has a resolution of 20-km at nadir, while the ABI pixel has 2-km resolution. By linear interpolating spatially the CERES fluxes across the ABI pixel does not properly distribute spatially the CERES flux observation (by not preserving cloud edges) and I would not consider that a truth dataset, since it does not represent the observed 2-km fluxes, It would be better to map the ABI pixels into the CERES footprint to validate the NTB algorithm. A CERES footprint at 60° view angle (near the scan edge) has a 40-km extent encompassing over 400 ABI pixels at nadir. Even better would be to evaluate the ABI product regionally, say for 1° regions, so that monthly regional comparisons can be made.

Fig 10 caption missing (e)

Line 326 Based on Table 6, the ABI radiances, aerosols, cloud mask, phase and optical depth are used as inputs. For clear-sky the surface spectral reflectance is based on 12 IGBP types, and 4 types for cloudy types. How is the pixel level above surface or cloud top amount to account for NIR atmospheric water vapor absorption. A lot of effort was used to define atmospheric profiles, I would assume this would be based on the ABI channel radiances. My other concern is that the 0.86 vegetation reflection is a function of season and region, in winter the leaves have fallen off the trees, where as in summer the trees have leaves. By simply relying on IGBP type does not account for the seasonal vegetation reflection.

Line 346 and line 33. Given that the ABI sampling is less than 15 minutes. The 7.5 minute difference is very small. Once the SW fluxes are compared at the footprint or regional scales the

time difference will not make much of a difference in the bias. All Terra and Aqua overpasses should be matched for well sampled validation results. The following paper Fig. 2 shows that the time difference does not dramatically increase the matching noise
B. A. Wielicki, D. R. Doelling, D. F. Young, N. G. Loeb, D. P. Garber and D. G. MacDonnell, "Climate Quality Broadband and Narrowband Solar Reflected Radiance Calibration Between Sensors in Orbit," IGARSS 2008 - 2008 IEEE International Geoscience and Remote Sensing Symposium, 2008, pp. I-257-I-260, doi: 10.1109/IGARSS.2008.4778842

Line 348 I agree that seasonal/regional variation of the NIR vegetation reflection must be taken into account.

Line 358 The CERES edition 4 ADMs also rely on NDVI, which accounts for changes in the vegetation NIR reflectance. The CERES edition 2 relies on surface types only.

Su, W., Corbett, J., Eitzen, Z., and Liang, L.: Next-generation angular distribution models for top-of-atmosphere radiative flux calculation from CERES instruments: methodology, Atmos. Meas. Tech., 8, 611–632, https://doi.org/10.5194/amt-8-611-2015, 2015

Line 38- what is the source of the open shrub, desert, woody savanna and grassland spectral albedos? Are these TOA albedos?

Line 397. I agree there is no truth dataset for 2km resolution BB fluxes. That is the reason why this dataset is being produced. In order to perform a fair comparison, the high resolution ABI pixels fluxes must be mapped into the CERES footprint, or both reduced to a 100-km region in order to track the ABI and CERES over the record.

Line 405. "transformation of narrowband quantities into broadband ones" This sentence is ambiguous.

Line 414. What is this sentence trying to say? "The process of preparing for the usefulness of a new satellite sensor needs to be done in advance, the final configuration of the instrument becomes known at a much later stage." This was not addressed in the paper

Line 416 What is this sentence trying to say? "As such, the evaluation of the new algorithms is in a fluid stage for a long time." Usually there is an initial release and as the algorithms improve incrementally while the version number is updated over time. For example MODIS L1b C6.1 dataset is currently available and C7 is being developed and tested.

Line 417 This sentence is confusing. "Agreement or disagreement with know "ground truth" is not fully informative on the performance of the new algorithms to estimate desired geophysical parameters." Are you talking about compensating errors?

Line 420 reliable cloud screening and cloud properties. What about non-retrieved cloud properties from cloud mask identified pixels? What about optically very thin clouds where the surface contributes to the TOA reflectances.

Line 421 The CERES SSF L2 Edition 4 product SW fluxes has been available prior to ABI and have not gone through any major revisions. The SSF1deg fluxes have been used to monitor global and regional SW flux variability over time. On the other hand the FlashFLUX has undergone revisions.

What is the application of high-resolution ABI TOA SW fluxes, that the low resolution CERES fluxes cannot fulfill? For the application, what is the required SW TOA accuracy?

Line 429 If the ABI aerosol algorithm does not ever reach stability in the future, will the TOA SW product ever be released?

---

## Editor Comment (EC1)

Comment for manuscript amt-2021-289 on behalf of one of the reviewers.

Dear authors,
one of the reviewers communicated with me and sent some follow-up comments regarding the revised version of your manuscript. I post them here in the public discussion because they seem appropriate for the manuscript's public record. I encourage you to post a point-by-point response when making edits for the final version of the paper.
Thank you,
Sebastian Schmidt (editor)

Comment by reviewer in response to the revised version, and also in response to AC3 (https://doi.org/10.5194/amt-2021-289-AC3)

The authors could have done more to address my initial feedback, addressing the following clarification comments would be greatly appreciated.

It is still unclear to me if the paper is a validation paper of the NOAA STAR TOA SW flux product and if so the dataset and version number should be properly cited. If the GOES SW TOA flux product is being produced by NOAA it should be cited. If it is not, then it should also be stated in the text. If this is an algorithm paper of a potential NOAA product that is in development that should be clearly stated.

3.1 Satellite data for GOES-16 and GOES17: datasets are used in papers I expect the product name, version number and location should be given. I find section 3.1 completely lacking in this regard. First of all, I searched for https://www.bou.class.noaa.gov/ and the site could not be found. I do not know if this is the GOES L1b radiance data, since the product name was not given in the text. The text mentions that "The CODC data are not always available from CLASS". Could the authors provide the name and version of the product of the cloud retrievals used in this study. Lastly the GOES based TOA flux dataset or product promoted in this paper is not cited in the paper.

3.2 Reference data from CERES: This section is completely confusing. Some of the figures were used from CERES SSF L2 and for fig. 9 the CERES FLASHflux level 2. Again, the edition numbers were not cited. I believe it was CERES SSF L2 Edition4 and FLASHflux Version 4A. This is extremely important if someone wanted to recreate the results in the future when the CERES project may have moved on to Edition 5.

I was disappointed that only a few overpasses were validated in the paper and here is the response from the authors. "The ABI is at 5 min intervals. However, we want to compare four products simultaneously. It is hard to find cases when all of the GOES-16, GOES-17, CERES/Terra and CERES/Aqua have overlap in time and that the overlap is large enough to compare all of them." For me, there is no stipulation that they need to be validated simultaneously in order to have a robust validation matched dataset.

I looked at the ESMF re-gridding web site, there are multiple grid type options. Could the gridding algorithm just be simply detailed in the text.

The point of the paper is that the CERES and GOES surface types could be a factor. The Su et al. 2015 ADM type are more a function of NDVI over land and not strictly dependent on IGBP type and that NDVI allows for seasonal variability, whereas the GOES (this paper) has a static surface type categories not allowing for seasonal variation of interannual variability.

Line 389. The "ground truth", namely, the CERES observations are also undergoing adjustments and recalibration, is misleading. The CERES SSF L2 TOA flux observations have been using consistent algorithms and instrument calibration across a CERES edition (not FLASHflux). That is a new edition is reprocessed from the beginning of record with consistent algorithms and calibration. That is why citing datasets is so important.

In the abstract the last sentence states: A satisfactory agreement between the fluxes was observed for both clear and cloudy conditions and possible reasons for differences have been identified." Satisfactory agreement is a relative term. I believe that the authors need to describe who their users are and that the level of agreement is sufficient for their applications.

---

## Author Comment (AC1)

**Response to Reviews                                    12/10/2021**

**Color Coding:**
Black: Reviewer
Red: Authors

We thank this Reviewer for very helpful comments.

**Comment:** There is some displacement in the line numbering of the Reviewers and the submitted version of the manuscript. We have also downloaded the ATM version, but there is still a shift. This made it sometimes difficult to identify the location of the Reviewers comments.

**Reviewer # 3**

**RC3**: Comment on amt-2021-289', Anonymous Referee #3, 05 Nov 2021 reply
Review for "Top of the atmosphere reflected shortwave radiative fluxes from GOES-R" by Pinker et al.

This paper described the methodology developed to derive surface and TOA SW radiative flux from ABI onboard GOES-R. It includes the conversion of the narrowband radiance observations from ABI to broadband SW radiances that are needed, and the subsequent conversion of broadband SW radiances to broadband SW fluxes. Authors used the MODTRAN to derive the narrowband-to-broadband regression coefficients first for each of the 6 channels and then used the weighed sum for the final SW broadband reflectance. These broadband radiances were then converted to fluxes using a hybrid ADMs from CERES observations and MODTRAN simulations. Major concerns are:

**Reviewer # 3**

1.  For the narrowband-to-broadband conversion, the best strategy would be to use common channels on ABI and MODIS (VIIRS) and then develop the regressions using CERES Level 2 SSF data where CERES broadband radiances and MODIS (VIIRS) narrowband radiances are collocated. Spectral band difference adjustment factors (Scarino et al., 2016) can be used to account for the SRF differences between ABI and MODIS (VIIRS). I also recommend using the multi-linear regressions instead of the two-step approach used here.

**Response**
This is an interesting option and could serve as a new future project. It is feasible when the CERES and VIIRS are on the same satellite as is the case for CERES and VIIRS for S-NPP and NOAA-20. However, our approach is feasible for any combination of satellites. This comment highlights the difficulty one faces when preparing for a new instrument, in this case, ABI. Since the period of stabilization with the instrument at hand is relatively long, independent developments are not at stand still. To catch-up with all of them would be possible only as a new effort. We believe that

what was done is of interest and opens new possibilities for future investigations. We have initially used multi-linear regression but encountered problems of getting negative radiance values. The approach used ensured that all values were positive.

Many operational algorithms, including the NOAA STAR SRB algorithm, are required to work reasonably well on day one, that is, right after the launch of the satellite. The data needed for the approach the reviewer is suggesting are not available from the new satellite at that time and would need to be collected for a long enough period to include all seasons. So, simulation-based narrow-to-broadband transformations are necessary for the day-one algorithm to work. Having said that, we note that an empirical narrow-to-broadband conversion using coincident and collocated CERES and ABI observations is being developed at NOAA/STAR and the method is briefly mentioned in Laszlo et al. (2020).

Reviewer # 3

2. The CERES ADMs that the authors used in the study is outdated. I believe those ADMs are based on the CERES on TRMM observations, as the justification that you used to calculate theoretical ADMs is because "CERES observations at higher latitudes are under-sample or not existent". The ADMs from Loeb et al. (2005) and Su et al. (2015) are based on Terra and Aqua observations and provide sufficient coverage over high-latitude regions. The methodology that you developed to combine the CERES and theoretical ADMs are thus not necessary.

**Response**

We have used the CERES ADMs that were mature at the start of the project. Indeed, it would be interesting to do what this Reviewer is proposing and compare the results to what was done under this project. Independent approaches are useful to identify issues with any path one may take.

Still, we were looking for the new ADMs however, unlike the CERES/TRMM SSF Edition 2B ADMs, the more recent CERES Terra/Aqua Edition4 ADMs are not available publicly, at least we could not locate them at the NASA LaRC website https://ceres.larc.nasa.gov/data/angular-distribution-models/.

Reviewer # 3

3. As authors mentioned in this paper, CERES provides TOA SW fluxes, it is not clear from the manuscript why fluxes from ABI are necessary. What are the objectives for deriving fluxes from ABI and what are the potential applications?

**Response**

Development of the NOAA shortwave radiation products (reflected shortwave radiation at TOA and downward shortwave radiation at the surface) was performed in response to a requirement of the NOAA National Weather Service (NWS) for such a product. The Environmental Monitoring Center of NWS requested this product for verifying model-predicted shortwave radiation to improve radiation and land-atmosphere interaction processes (Laszlo et al., 2020).

**Specific comments:**

Reviewer # 3
1. Line 28, "A satisfactory agreement between the fluxes…" is very vague, including biases and RMS errors will be helpful.

Response
This is mentioned in the Abstract which is limited in length. Such information is provided later in Table 7.

Reviewer # 3
2. CERES ADMs are scene specific, the flowchart in Fig. 2 indicates that cloud phase and cloud optical depth are used for ADM. However, the paper didn't describe how these cloud properties are derived.

Response
We did not derive these cloud properties. We use the same classification as used in CERES to make the allocation of the ADMs consistent. We believe that it is out of the scope of this paper to describe the monumental work of CERES.

Reviewer # 3
Line 116, "The difference between the two radiances were below 5%", is the difference for broadband radiances or any specific wavelength?

Response
It was done spectrally for the entire spectrum.

Reviewer # 3
one should avoid using red and green color scheme.

Response
Thanks for pointing this out. We will be careful about it in the future.

Reviewer # 3
3. Line 194, wrong figure number.

Response
Corrected.

Reviewer # 3
4. , it is hard to see the gray lines.

Response
Will attempt to improve.

**Reviewer # 3**

5. didn't separate the comparison into clear versus cloudy conditions, but authors mentioned on line 244 that "The separate-channel" coefficients work well for predominantly clear sky". I assume authors draw this conclusion based upon the flux magnitude rather than any cloud detection algorithm? Magnitude of TOA SW flux is smaller under clear-sky conditions than under cloudy-sky conditions. Absolute flux differences are not the best way to assess the performance for clear- and cloudy-sky conditions.

**Response**

In respect to cloud mask, we used the official ABI cloud mask. The other comment of this Reviewer is well taken. We have redone the analysis for clear and cloudy sky and computed biases and RMSE. We have prepared the following new Figure to illustrate it. Indeed, this is a better way to look at the differences. As seen now, difficult to reach a conclusion about performance under clear and cloudy sky. Thank you for this comment.

[Figure]

Figure. The relative statistics for Bias and RMSE. The y-axis is percentage. The x-axis are the cases used for comparison. The blue color is for cloudy conditions, the orange is for clear sky and the gray is for the all sky cases.

**Reviewer # 3**

6. Why using CERES FLASHFLUX for validation? I understand the latency issue, but the data presented in this study are from 2017. Surely higher quality CERES (i.e., SSF) are available now for 2017.

**Response**

We agree with this Reviewer on the need to use the final CERES product. Indeed, that is what was done in all the cases described in the paper. We apologize for the mistake we made in labeling the product. We have been involved in using the FLASHFlux data in preliminary evaluation for such a long time (due to the circumstances of data availability) that we labeled it as Flash Flux and not CERES. We have now prepared a full data base of what was used so the reader can check this point out. Information will be provided how to access this database.

**Reviewer # 3**

7. CERES data are of much coarse resolution (~20 km) compares to that of ABI (~2 km), the spatial resolution differences will certainly contribute to the biases and RMS. Authors should consider revise the comparison method by averaging the ABI pixels within the CERES footprints weighted by the CERES point-spread function before comparing with the CERES flux.

**Response**

For the re-mapping, we adopted the ESMF re-gridding package. The detailed information can be found at:

http://earthsystemmodeling.org/regrid/

For an ideal situation, the ABI high-resolution TOA SW fluxes should be mapped into the CERES footprint for validation as suggested by the Reviewer. However, there are reasons that make it difficult to do so. For example, the case 12/26/2019 UTC 19. There can be more than 18000 pixels in a single swath of the SSF, when constrained to U.S. Different pixels have different times. Neglecting the seconds, there are still more than 30 mins differences (this changes case by case) between the first pixel and the one at the end and this brings up a time matching time issue. But if remapping the SSF to ABI, we can set up a unique time for ABI (ABI is at 5 min intervals) and then constrain the region and the time range of SSF.

Both remapping the ABI to SSF and remapping SSF to the ABI bring up spatial matching errors as recognized by the scientific community. In Figure 10, we show the SSF before re-gridding (Figs 10 (a) & (b)) and after re-gridding (Figs. 10 (c) and (d)). As seen, the fluxes after re-mapping CERES SSF to the ABI resolution resemble well the reverse re-mapping. Another consideration is the computational efficiency of re-mapping the curvilinear tripolar grid to unconstructed grid. For large arrays, it is more efficient to remap the unconstructed grid to the curvilinear tripolar grid.

We have done one case of remapping the ABI to CERES_SSF as suggested, and the edges do improve. There are additional consideration in selecting the direction of re-mapping. This Figure will be put in the Supplements.

**Remapping 2-km ABI flux to CERES_SSF scale (20 km)**

[Figure]

FigureS1. Top Left: Mean ABI Flux on 12/26/2019 UTC 19:00 from GOES-16 re-gridded to CERES SSF (20km)/Aqua domain; Top Right: Difference between re-gridded ABI Flux and CERES SSF/Aqua; Bottom: frequency distribution of the differences (bottom).

Reviewer # 3

8. Line 256, what "CODC" stands for?

Response
Cloud Optical Depth Conus

Reviewer # 3

9. Line 271, typo.

Response

Corrected.

Reviewer # 3

10. Line 321, authors state that "both estimates of TOA fluxes do no(t) account for seasonality in the land use classification", this is not clear. Do you mean CERES ADMs do not account for land surface seasonality? If so, that is not true. CERES clear-land ADMs are constructed for each calendar month (Loeb et al. 2005, Su et al. 2015).

Response
The simulations do not account for seasonality.

Reviewer # 3

11. Line 376, what do you mean "the order in which these transformations are executed is arbitrary"?

Response
Was reworded.

Reviewer # 3

12. Line 388-389, CERES Ed4 data were release in 2017 or so, not sure what authors mean that "CERES observations are also undergoing adjustment and recalibration". Please clarify.

Response
Was clarified in text. Thanks for the additional references that are now included in our paper.

Scarino et al. (2016), A Web-Based Tool for Calculating Spectral Band Difference Adjustment Factors Derived from SCIAMACHY Hyperspectral Data, IEEE Trans. Geo. Remote Sens., 54, 5, 10.1109/TGRS.2015.2502904.
Su et al. (2015), Next-generation angular distribution models for top-of- atmosphere radiative flux calculation from the CERES instruments: Methodology. Atmos. Meas. Tech., 8:611–632.
Loeb et al. (2005), Angular distribution models for top-of- atmosphere radiative flux estimation from the Clouds and the Earth's Radiant Energy System Instrument on the Terra satellite. part I: Methodology. J. Atmos. Oceanic Technol., 22:338–351.

---

## Author Comment (AC2)

**Response to Reviews                                    12/10/2021**

**Color Coding:**

Black: Reviewer

Red: Authors

We thank this Reviewer for very helpful comments.

**Comment:** There is some displacement in the line numbering of the Reviewers and both the submitted and the ATM versions. This made it sometimes difficult to identify the location of the Reviewers comments.

**Reviewer # 1**
Review of "Top of the Atmosphere Reflected Shortwave Radiative Fluxes from GOES-R" by Pinker et al.

21 October 2021

**Overview**

The manuscript prepared by Pinker et al describes the conversion of radiances from the ABI instrument on GOES-R to SW radiative fluxes. First, a spectral regression is applied to convert narrow band radiances to broadband radiances. Second, angular distribution models are applied to convert the broadband radiances to radiative fluxes. The derived radiative fluxes are compared to those from the CERES FLASHFlux product. Possible reasons for discrepancies are discussed.

This work addresses an important and interesting topic, and I believe that SW radiative fluxes from GOES have the potential to be of great value to the scientific community. However, I have several major concerns as outlined below. In summary, there are significant gaps in the description of the methods that need addressing, and the reasons for differences with CERES data would benefit from some additional analysis. After addressing these concerns, I believe the work would be a good fit for publication in Atmospheric Measurement Techniques.

**Major comments**

**Reviewer 1**

L99: In order to apply equation 3, there is an assumption that ADMs from observations and simulations for a given scene type belong to the same population. I am not convinced this is the case. If the CERES anisotropic factors and the simulated anisotropic factors are substantially different (e.g. due to neglected processes in the simulations such as 3D radiative effects), the weighted average anisotropic factor from equation 3 might end up somewhere in the middle, not representing either. I suggest discussing this caveat, or addressing this issue with a figure

showing that the underlying radiances for a challenging scene type largely overlap between the simulations and CERES.

**Authors**

The comments of this Reviewer are well taken. We were also concerned about the same issues. We have done numerous experiments to understand the sources of differences between the theoretical and CERES ADMs to convince ourselves that the synthesis of the two is sound, even if the two approaches are not identical. In **Figure 1** the patterns of bi-directional correction differences for desert under clear-sky from MODTRAN simulations and CERES observations are illustrated. Largest difference occurs for higher VZAs. While inaccuracies in the specific surface spectral reflectance used in the simulations may contribute to the differences, our experiments show that they are most likely due to differences in sampling frequency of observations at high VZAs. A hybrid approach is applied that hopefully is compensating for the uneven-sampling in the two methods.

[Figure]

Correction Factor

*Figure 1.     Bi-directional correction factors at SZA 63.2° over desert for clear-sky*
*Left: Simulations; Right: CERES observations (Bright Desert)*

Before undertaking the simulations, we had to develop a method to reconcile different scene types and angular binning of the CERES and simulated ADMs and a weighting function to combine the two data sources. CERES-TRMM clear-sky ADM classification by surface types does not fully match the IGBP surface classification. In the simulations, the 12 IGBP surface classifications are used. For clear sky, there are 8 surface types in CERES ADMs. An effort was made to combine the corresponding CERES ADMs and simulated ADMs based on IGBP scene classifications to generate new synthesized ADMs for 12 IGBP surface types. The cloud classification in CERES ADMs is based on Cloud Optical Depth (COD) and cloud phase (water cloud, ice cloud) over ocean, low-mod tree/shrub, mod-high tree/shrub, desert, and snow/ice.

For clear sky, the synthesized ADMs are generated from a combination of simulated and CERES bi-directional correction factors based on IGBP surface classifications for each angular bin by weighting, as presented in the manuscript. For example, CERES Low-Mod Tree/Shrub ADMs are grouped from observations of the following three IGBP surface scenes: Savannas, Grassland, and Crops/Mosaic (Loeb et al., 2003). The difference in the bi-directional correction

factors between the combined and CERES ADMs for Savannas is shown in **Figure 2.** At lower viewing zenith angles the percentage of differences is mostly within +/- 10% but the differences are much larger at higher viewing zenith angles.

[Figure]

*Figure 2.*     *Distribution patterns of the difference of the bi-directional correction factor between combined ADMs and CERES ADMs for Savannas over clear sky at Solar Zenith Angle of 70-80°:*
*Left:    Difference (Combined ADMs – CERES ADMs)*
*Right:  Percentage of Difference (Difference/CERES ADMs)*

At an early stage of this work when ABI observations were not yet available, we have tested the approach with SEVIRI observations. The following Table (Niu and Pinker, 2011) illustrates that using the hybrid approach results in better agreement with CERES compared to what was achievable with CERS ADMs alone.

Table 7. Evaluations of July 2004 monthly mean TOA upward SW flux estimates as driven with SEVIRI observations when using CERES ADMs or synthesized ADMs, against CERES observations (SRBAVG product).

| | BIAS (W m$^{-2}$) | | RMSE (W m$^{-2}$) | |
| --- | --- | --- | --- | --- |
| Statistical results | CERES ADMs | Synthesized ADMs | CERES ADMs | Synthesized ADMs |
| Clear sky (7801 samples) | 8.7 | 4.6 | 7.1 | 6.5 |
| All sky (8128 samples) | −3.1 | −2.7 | 8.2 | 6.3 |

As mentioned in our manuscript, we have originally prepared two papers. The first one summarized the early results with proxy observations like SEVIRI, GERB, MODIS etc. where some of these issues are explained in detail. Due to concern that the early material may not be any more of interest to the readers, we have focused in the second paper on ABI using the latest versions of GOES-16 and 17 data.

**Reviewer 1**
L103: How is it possible to know "m", ie. the number of CERES observations associated with the anisotropic factor for each angular bin? If I understand correctly, the authors are using the existing CERES ADMs derived from the CERES instrument on TRMM (Loeb et al., 2003), combined with their simulations. These CERES ADMs provide anisotropic factors but, to my knowledge, they do not provide the number of observations that were used to derive the anisotropic factors in each angular bin.

**Authors**
Please see response to previous comment.

**Reviewer 1**
L109: What is the "tool" that was developed to select 100 profiles from the original database of 15704? How does it ensure a variety of conditions are represented? Details are needed, otherwise there is no way that the results can be reproduced.

**Authors**
An effort was made to have representation of different climatic regions and covering all seasons equally. This selection depends on the availability of observations, namely, if less soundings are available in certain region, that region will be under-represented.

**Reviewer 1**
L164-171: Some key information is missing relating to how clouds are included in the simulations. The following should be included in Table 3:
- What is the phase of each cloud type? I assume cirrus is ice, stratocumulus is liquid. Altostratus is a mixture? What ice optical properties are used in the simulations?
- Are the 3 cloud types always simulated in isolation, or does the set of simulations include combinations ie. multi-layer cloud?
- Is there any attempt to consider cloud fraction?

**Authors**
The cloud model is the MODTRAN built-in one. The table below gives information from the MODTRAN manual. All clouds are assumed single layer type. Cloud fraction is considered in the flux calculation step. For N2B and ADM conversion, each pixel or field-of-view is assumed to be either clear or total cloudy.

Properties of the MODTRAN Cumulus and Stratus Type Model Clouds.

| ICLD | Cloud Type | Thickness (km) | Base (km) | .55µm Ext. (km$^{-1}$) | Column Amt. (km gm / m$^3$) |
|---|---|---|---|---|---|
| 1 | Cumulus | 2.34 | 0.66 | 92.6 | 1.6640 |
| 2 | Altostratus | 0.60 | 2.40 | 128.1 | 0.3450 |
| 3 | Stratus | 0.67 | 0.33 | 56.9 | 0.2010 |
| 4 | Stratus/Stratocumulus | 1.34 | 0.66 | 38.7 | 0.2165 |
| 5 | Nimbostratus | 0.50 | 0.16 | 92.0 | 0.3460 |

**Reviewer 1**

L264-267: The differences shown in Fig 9c and 9d occur after applying a NTB conversion and then ADMs. The authors claim that the reason for the differences could be the temporal offset between CERES and GOES. I am not convinced. The observations are co-located to within 5min. Not many cloud regimes are drastically changing within 5min at the CERES footprint scale. I expect the uncertainty due to the NTB conversion and ADMs is much larger. For the NTB conversion in cloudy scenes, one possible reason is that the ABI bands do not provide sufficient spectral coverage. Figure 1b in Gristey et al., JClim, 2019

Et us discuss(https://doi.org/10.1175/JCLI-D-18-0815.1) shows SW spectral reflectance variations for different cloud types. Comparing with the ABI bands, I suspect some spectral variations associated with cloud variability are missed. For ADMs in cloudy scenes, the cloud properties must be retrieved for the selection of the correct ADM. Misclassification of cloud properties will therefore result in flux differences. Even if the correct scene type is selected, ADMs have an uncertainty due to within-scene variability and within-angular bin variability. I suggest including discussion of these possible reasons.

**Authors**
Thank you for pointing this out. We have now included these comments in the manuscript. We have stated:
As discussed in Gristey et al. (2019) there are SW spectral reflectance variations for different cloud types. Possibly, for ABI bands some spectral variations associated with cloud variability are missed. It is important to have the correct cloud properties to be able to select correct ADM. Misclassification of cloud properties will therefore result in flux differences. They also argue that ADMs have an uncertainty due to within-scene variability and within-angular bin variability leading to additional flux differences.

**Reviewer 1**
L271: This section does not mention the time range/case studies of observations used from GOES-16 and GEOS-17. Some cases are listed in Table 7, but this table is not referenced anywhere in the text. It is not clear if these cases studies encompass all of the data used in the study. Again, this is essential information for anyone interested to reproduce the results

**Authors**

These are all the cases that have been re-done against the updated CERES data. Numerous other cases were compared with the FLASHFlux data at an earlier stage. We now reference Table 7 in the text and we add information on the time range in this table.

**Reviewer 1**

L293-294: There seems to be an inconsistency here. The previous paragraph states that FLASHFlux was used because the GOES data was only available for about a week, and FLASHFlux is available within that timeframe. Fair enough. But then it is stated that GOES data is now available in the CLASS archive going back to 2017. So, there is no longer a valid reason to perform comparisons against the (less accurate) FLASHFlux data. Is there any reason that the authors cannot perform their analysis using the GOES data from the CLASS archive against the primary CERES L2 SSF product? Maybe I am missing something.

**Authors**

No. This Reviewer is not missing anything. Indeed, there was some confusion regarding what was used in the latest version of comparison as presented in Table 7. We apologize for the mistake we made in labeling the product. We have been involved in using the FLASHFlux data in preliminary evaluation for such a long time (due to the latency of data availability) that we labeled it as Flash Flux and not CERES. We have now prepared a full data base of what was used so the reader can have a hold of the data used. Information will be provided how to access this database.

**Reviewer 1**

L296: A major step missing from the paper is how the scene properties are determined for the ABI observations. I expected to see details in this section. My understanding is that both the regression coefficients for the NTB conversion and the ADMs are a function of scene type. I see that a fixed surface type is assumed but how are the changing atmospheric properties accounted for when converting the ABI narrow band radiances to broadband fluxes?

**Authors**

We use the IGBP classification under clear conditions. Table 2 describes the surface classification for IGBP 18 types, and their reduction to 12 IGBP types as used in this study to match the CERES types. This is also discussed in the text. For cloudy conditions we select the N/B and ADMs according to cloud classification and optical depth.

**Reviewer 1**

L298: I find it strange that the authors decided to perform their comparisons at the ABI spatial resolution by applying a bi-linear interpolation to the CERES data. It would make more sense to aggregate the ABI data and perform comparisons at the CERES footprint scale. By performing comparisons at the coarser of the two scales, non-linearity due to interpolation is not an issue.

**Authors**

For the re-mapping, we adopted the ESMF re-gridding package. The detailed information can be found at:

http://earthsystemmodeling.org/regrid/
For an ideal situation, the ABI high-resolution TOA SW fluxes should be mapped into the CERES footprint for validation as suggested by the Reviewer. However, there are reasons that make it difficult to do so. For example, the case 12/26/2019 UTC 19. There could be more than 18000 pixels in a single swath of the SSF if constrained to the region of U.S. Different pixels have different times. Neglecting the seconds, there are still more than 30 mins differences (this changes case by case) between the first pixel and the one at the end and this brings up a time matching issue. But if remapping the SSF to ABI, we can set up a unique time for ABI (ABI is at 5 min intervals) and then constrain the region and the time range of SSF.

Both remapping the ABI to SSF and re-mapping SSF to the ABI bring up spatial matching errors as recognized by the scientific community. In Figure 10, we show the SSF before re-gridding (Figs 10 (a) & (b)) and after re-gridding (Figs. 10 (c) and (d)). As seen, the fluxes after re-mapping CERES SSF to the ABI resolution resemble well the original CERES. A case of reverse mapping is shown in the Appendix and indeed as the Reviewer suggested, it reduces the edge effects. Another consideration is the computational efficiency of re-mapping the curvilinear tripolar grid to unconstructed grid. For large arrays, it is more efficient to remap the unconstructed grid to the curvilinear tripolar grid.

**Reviewer 1**
317: There is no reference to Fig 11, 12 or 13 in the text. These figures are key to the findings of the study and should be referred to throughout the results section.

**Authors**
References have been added to the text.

**Reviewer 1**
L358-365: I do not necessarily disagree with these comments on possible differences in the surface spectral reflectance, but they are purely speculative and insubstantial. Can any supporting analysis be added? For example, MODIS provides a surface spectral reflectance co-located with CERES on both Aqua and Terra, albeit at a coarse spectral resolution. The observed MODIS surface reflectance could be compared with MODTRAN values, even just for a handful of case studies, to quantify any differences.

**Authors**
We have removed now section 5.1.2 since it caused some concerns.

**Reviewer 1**
L390-400: Again, the text here relating to the temporal offset between GOES and CERES is speculative and would be much better served by some supporting analysis. I suggest including a scatter plot using the same data in Fig. 10. The x-axis would be the temporal offset (ranging from 0 to 5 min) and the y-axis would be the difference between GEOS and CERES. Data points could be colored by scene type. If the temporal offset is an important issue, expect to see a clear positive gradient.

Authors
The GOES data come in 5 min granule but do not provide a time stamp for each pixel.

**Minor comments**

Reviewer 1
L38-44: The first paragraph of the introduction does not really serve a purpose. It is irrelevant for the analysis and does not add much to the manuscript in my opinion. It could be removed.

Authors
Is removed now.

Reviewer 1
L51-52: There is a recent review paper on shortwave ADMs that could be cited here: Gristey et al., 2021, https://doi.org/10.3390/rs13132640.

Authors
Thank you. We have now referenced this paper later in the manuscript.

Reviewer 1
L81: Down arrow in the text is out of place and should be removed.

Authors
Cannot find this arrow.

Reviewer 1
L129: Are the "surface variables" also part of the SeeBor dataset, or added by the authors? Please clarify when the dataset is first introduced.

Authors
The surface variables are part of SeeBor but we did not use them. We used information from MODIS.

Reviewer 1
L130: Is the surface albedo a single broadband value? If so, how is this combined with the spectral surface albedo used in MODTRAN (discussed later).

Authors
Spectral albedo.

Reviewer 1
L132: There is a positive bias in what variable? At what altitude? Please be more specific. Fig. 4 shows 3 variables. The sign of the temperature bias depends on altitude; the water vapor bias is positive only at lower altitudes; the ozone bias is positive only at higher altitudes.

Authors
This is in reference to the temperature profiles. There is a positive bias at lower altitudes and negative bias above 1 mb. Was added to text.

Reviewer 1
L135: This section does not mention that the surface type is fixed in time. Implications are discussed later, but it should be stated clearly here since this is where the dataset is first introduced and it is an important aspect of the work.

Authors
Done.

Reviewer 1
L146: Under clear-sky, scattering by aerosol is important, but probably not multiple scattering. Most aerosol loadings are dominated by single scattering. Suggest removing "multiple".

Authors
Done.

Reviewer 1
L146: In addition to aerosol scattering, what about the role of absorption? The 6 aerosol types considered presumably have different single scatter albedo.

Authors
Build in the MODTRAN model. They represent aerosols with different single scatter albedos.

Reviewer 1
L157: Please provide an explanation of where the number 288,000 comes from. I calculated 6 aerosol types x 12 surface types x 100 profiles = 7200 simulations for clear-sky.

Authors
288,000 is wrong. It comes from 6 aerosol types x 100 profiles x 480 angles. However, the 480 angles do not include edge angles, that is, $0°$ for solar zenith angle, $0°$ and $180°$ for relative azimuth angle and $180°$ for satellite viewing angle. If including these edge angles, the number of MODTRAN simulations for each surface type is 462,000.

Reviewer 1
L162: How are the variations at 4 different wind speeds accounted for. The 100 profiles do not include wind speed information. I also assume this is surface wind speed but please clarify.

Authors
We do not use this option. Text modified.

Reviewer 1
L176: How is the number of Gaussian quadrature points determined? A sentence or two explaining the use of Gaussian quadrature would help the reader here.

Authors
The Gaussian angle cosines are equally spaced. The angles were previously selected for flux computations.

Reviewer 1
L182: "azimuth angle" should be "relative azimuth angle", I think.

Authors
Corrected to relative azimuth angle.

Reviewer 1
L184: "ignoring spherical geometry" – what does this mean?

Authors
The satellite zenith angle at the surface and at the TOA would be little different if considering the earth spherical curvature. For simplicity, we assume plane parallel geometry when converting satellite zenith angle from TOA value to surface value.

Reviewer 1
L226: 8-stream is used as the baseline/truth in Fig 7, but I do not see any evidence that 8-stream is itself sufficient. If the number of streams was further increased to eg. 16 or 32, would there be any benefit?

Authors
We used a scaled two stream RT solver to speed up the simulation. It is basically a two-stream scheme that is calibrated with 8-stream solution at a few spectral points. The accuracy benefit obtained from a higher number of streams may be totally lost in between the anchor points where a two-stream scheme is used. Also, we are mainly focusing on spectral conversion from narrow to board band, the impact of number of streams may be not that significant.

Reviewer 1
L231: Yes, the results for Scaled Isaacs are better than Isaacs, but how to quantify that they are "satisfactory"? I noticed that they are typically much worse than 4-stream DISORT in Fig 7b.

**Authors**
As illustrated in the Figure. We did not keep all the data so difficult now to get numerical values.

**Reviewer 1**
L260: Switching between wavelength and wavenumber is confusing for the reader. Since SW radiation is usually expressed in wavelength, and most of the plots in this study are in wavelength, I strongly suggest converting any instances of wavenumber throughout this manuscript to wavelength for consistency.

**Authors**
We have replotted Figure 8 in wavelength.

[Figure]

**Reviewer 1**
L264: "Figure 9" -> "Fig. 9" for consistency.

**Authors**
Corrected.

**Reviewer 1**
L292: "2019however" – needs fixing.

**Authors**
Done.

**Reviewer 1**

L301: "must be less than ±5 min". Is this threshold based on any analysis? What is special about 5 min?

**Authors**

The basic reason is that the GOES-R data (for the CONUS region) are provided in granules of 5 min interval. We set up a unique time of ABI and then constrain the region and the time range of CERES SSF.

**Reviewer 1**

L312: At the footprint scale of which instrument, CERES or ABI?

**Authors**

CERES.

**Reviewer 1**

L321-324: This text is a repeat of the previous section and is not needed again here.

**Authors**

We could not find the line number but we have modified the text so perhaps it is gone.

**Reviewer 1**

L354: Where is the section 5 heading? It jumps straight from section 4.1 to section 5.1. Are there other subsections from section 4 that are missing?

**Authors**

We have modified the section numbering so this problem is gone.

**Reviewer 1**

L377: "the calculated broad-band reflectance was around 0.45" – was this for cloud free scenes only?

**Authors**

Yes.

**Reviewer 1**

L379: Agreed that the filter function for channel 6 (Fig 14) could be problematic. But what impact does this have on the total NTB conversion? What is the weight associated with channel 6?

**Authors**

The solar irradiance in the spectral interval assigned to channel 6 is 87 w/m2. Included in Figure 8 discussion.

**Reviewer 1**
Table 1: The first part of the caption is not necessary.

**Authors**
Removed.

**Reviewer 1**
Table 1: ABI band 3 is NIR, not VIS.

**Authors**
Corrected.

**Reviewer 1**
Table 1, column 2: "Central wavelength" would be better than "Channel". Need to include units.

**Authors**
Done.

**Reviewer 1**
Table 1, column 3: Are these spectral band widths associated with a threshold percent drop off in response?

**Authors**
It is explained in "https://www.goes-r.gov/spacesegment/ABI-tech-summary.html", *"TABLE I. Summary of the wavelength, resolution, and sample use and heritage instrument(s) of the ABI bands. The minimum and maximum wavelength range represent the full width at half maximum (FWHM or 50%) points. [The Instantaneous Geometric Field of View (IGFOV).]"*

**Reviewer 1**
Table 2: Could be more reader friendly. I suggest ordering the first column so that the groups in the second column are next to each other.

**Authors**
Done.

Table 2. Surface classification description for IGBP 18 types, IGBP 12 types, CERES clear sky 6 types, and NTB cloudy sky 4 types

| IGBP (18 types) | IGBP (12 types) | CERES clear-sky (6 types) | NTB cloudy-sky (4 types) |
|---|---|---|---|
| Evergreen Needleleaf | Needleleaf Forest | Mod-High Tree/Shrub | Land |
| Deciduous Needleleaf | | | |
| Evergreen Broadleaf | Broadleaf Forest | | |
| Deciduous Broadleaf | | | |
| Mixed Forest | Mixed Forest | | |
| Closed Shrublands | Closed Shrub | | |
| Woody Savannas | Woody Savannas | | |
| Savannas | Savannas | Low-Mod Tree/Shrub | |
| Grasslands | Grasslands | | |
| Permanent Wetlands | | | |
| Tundra | | | |
| Croplands | Croplands | | |
| Open Shrublands | Open Shrub | | |
| Urban and Built-up | Open Shrub | Dark Desert | Desert |
| Bare Soil and Rocks | Barren and Desert | Bright Desert | |
| Snow and Ice | Snow and Ice | Snow and Ice | Snow and Ice |
| Water Bodies | Ocean | Ocean | Water |

**Reviewer 1**
Table 4: "Azimuth angle" -> "Relative azimuth angle".

**Authors**
Relative azimuth angle".

**Reviewer 1**
Table 6: Not referenced anywhere in the text. I do not think it serves a purpose. Suggest removing it.

**Authors**
We reference it now.

**Reviewer 1**
Table 7: Not referenced anywhere in the text. List of dates and statistics are useful. I suggest keeping the table but making reference to it in the data/results sections.

**Authors**
Done.

**Reviewer 1**
Fig 1, box 2: "watervapor" -> "water vapor".

**Authors**
Thanks. Done.

**Reviewer 1**
Fig 1: Remove arrow leaving bottom box.

**Authors**
Done.

**Reviewer 1**
Fig 2: Remove floating arrow leaving the left of the first box.

**Authors**
We did not see it in our version.

**Reviewer 1**
Fig 3: End of caption is missing.

**Authors**
The arrow is indeed there in paper_2_GOES_R_ 05_27_2021_revised_08_05_2021.docx.
It is not in the latest version that will be submitted with the responses, so it is good.

**Reviewer 1**
Fig 3: Top of figure seems to be cut off.

**Authors**
Fixed now.

**Reviewer 1**
Fig 4: Suggest removing "(logarithmic scale)" from the caption. The error bars are plotted on the same (linear) scale.

**Authors**
Done.

**Reviewer 1**
Fig 6: Are these nadir radiances at TOA? What is the scene type? Need to include this information in the caption.

**Authors**
It is nadir.

**Reviewer 1**

Fig 7: Wavelength is increasing from right to left, opposite to the previous figure. For consistency, I suggest reproducing this figure with the wavelength increasing from left to right.

**Authors**

We opted to leave it as is.

**Reviewer 1**

Fig. 8: This figure is in wavenumber but others are in wavelength. For consistency, I suggest reproducing this figure in wavelength. Wavenumber could always be included as a second axis along the top of the plot.

**Authors**

It was re-plotted.

**Reviewer 1**

Fig 10: Labels need correcting in the caption. (e) is missing, (d) is in the wrong place.

**Authors**

Done.

---

## Author Comment (AC3)

**Response to Reviews                                    12/10/2021**

**Color Coding:**

Black: Reviewer

Red: Authors

We thank this Reviewer for very helpful comments.

**Comment:** There is some displacement in the line numbering of the Reviewers and the submitted version of the manuscript. We have also downloaded the ATM version, but there is still a shift. This made it sometimes difficult to identify the location of the Reviewers comments.

**Reviewer # 2**

**Suggestions for technical corrections or reasons for rejection**

I believe it is important to document satellite retrieval products so that the user can cite the product as well as understand it. This paper documents the NOAA STAR GOES ABI TOA SW flux product. The methodology section reads more like an ATBD, there are many tables on how the LUT parameters were binned. There is no science reasoning on why the algorithm was built the way it was. It simply jumps straight into the atmospheric profiles, aerosol inputs, etc., used in the radiative transfer model. For an algorithm paper, I expect that the validation to be based on the product data and I am unsure if the product exists. In this case, ABI channel data is utilized and processed through the algorithm. The authors should be using the official CERES SSF L2 product not the CERES FLASHFlux L2 product for validation. The FLASHFlux product was designed for real-time processing, where many inputs were replaced with real-time datasets, such as FPIT rather than GEOS.5.4.1 atmospheric profile data. Also, the FLASHFlux fluxes are not properly calibrated. Lastly the validation is based on a single swath of Terra and Aqua CERES fluxes. At least 4 seasonal months should be compared. I am also taken back, that by changing the surface type classification over Mexico the TOA SW flux can differ by 250 Wm-2, which is very large. CERES would help resolve these issues, but this case was not compared with CERES data. Unless I did not properly perceive the objectives of the paper, the paper is insufficient and incomplete to properly document the NOAA STAR GOES ABI TOA SW flux product and should be resubmitted.

**Response to Reviewer # 2**

**Reviewer # 2**

The methodology section reads more like an ATBD, there are many tables on how the LUT parameters were binned.

**Response**

The paper describes a methodology how to derive the TOA fluxes. Due to its nature, detailed tables are appropriate and similarity to an ATBD is unavoidable.

**Reviewer # 2**

There is no science reasoning on why the algorithm was built the way it was. It simply jumps straight into the atmospheric profiles, aerosol inputs, etc., used in the radiative transfer model.

**Response**

No new theory has been developed to deal with this problem. The procedures are straightforward as followed by other investigators that deal with this problem. The new aspect of the work is the implementation for a new satellite. The readers should be informed as how this step was implemented in the relevant inference schemes.

**Reviewer # 2**

For an algorithm paper, I expect that the validation to be based on the product data and I am unsure if the product exists. In this case, ABI channel data is utilized and processed through the algorithm.

**Response**

A product of shortwave fluxes does exist at NOAA/STAR. Our paper describes one part of the algorithm and as such, the validation is focused on this part. The evaluation of the entire algorithm that leads to the final product is out of the scope of this paper. There are many other issues involved in the implementation that can cause discrepancies at the surface. Evaluation against the SW at the surface would not tell the whole story about the quality of the TOA part.

**Reviewer # 2**

The authors should be using the official CERES SSF L2 product not the CERES FLASHFlux L2 product for validation. The FLASHFlux product was designed for real-time processing, where many inputs were replaced with real-time datasets, such as FPIT rather than GEOS.5.4.1 atmospheric profile data. Also, the FLASHFlux fluxes are not properly calibrated.

**Response**

We agree with this Reviewer on the need to use the final CERES product. Indeed, that is what was done in all the cases described in the paper. We apologize for the mistake we made in labeling the product. We have been involved in using the FLASHFlux data in preliminary evaluation for such a long time (due to the latency in data availability) that FLASHFlux was engrained in our memory. We have now prepared a data base of what was used so the reader can check this point out. Information will be provided how to access this database.

**Reviewer # 2**

Lastly the validation is based on a single swath of Terra and Aqua CERES fluxes. At least 4 seasonal months should be compared.

**Response**
We have added a case for summer so all the seasons are represented.
Factors that affect the NTB and ADM conversion include geometry angles, surface type and cloud optical depth for cloudy case. The surface type may be season dependent. We did compared cases in both summer and winter seasons. There are no significant differences in the statistics as seen in the new augmented Table 7.

**Reviewer # 2**
I am also taken back, that by changing the surface type classification over Mexico the TOA SW flux can differ by 250 Wm-2, which is very large. CERES would help resolve these issues, but this case was not compared with CERES data.

**Response**
Seems that this comparison is raising some questions. Since it is based only on one case, we have decided to eliminate this section.

**Reviewer # 2**
Unless I did not properly perceive the objectives of the paper, the paper is insufficient and incomplete to properly document the NOAA STAR GOES ABI TOA SW flux product and should be resubmitted.

**Response**
We thank you for this recommendation. We have done additional documentation of the work that was done, added more cases and we plan to resubmit the manuscript if our responses are found to be satisfactory.

**Supplementary Comments of Reviewer # 2**

**Reviewer # 2**

**General comments.**
The authors should be using the official CERES SSF L2 product not the CERES FLASHFlux L2 product for validation. The FLASHFlux product was designed for real-time processing, where many SSF inputs were replaced with real-time datasets, for example the GEOS 5.4.1 reanalysis rdataset was replace by the realtime FPIT dataset. The more realtime dataset algorithm datasets are often revised due to changing input quality. The CERES input datasets were designed for consistency across the record by limiting algorithm changes to avoid discontinuities in the parameter values. Also, the FLASHFlux fluxes do not employ the most up to date CERES instrument calibration coefficients.

Response

We have responded to this comment earlier. Indeed, that is what was done (mistake in labeling).

**Reviewer # 2**

I am assuming that the NOAA STAR GOES ABI TOA SW flux product is not available to the public.

Response
The comment of this Reviewer made us realize that perhaps, it should be made available to the public. Under consideration.

**Reviewer # 2**

I am having a hard time understanding why there is very little validation being performed. Table 7 simply is not sufficient, not even a full year of data is analyzed. It seems that exact time matching is necessary for validation. The ABI scans are every 15/10 minutes providing closely matched ABI and CERES fluxes.

Response
The ABI is at 5 min intervals. However, we want to compare four products simultaneously. It is hard to find cases when all of the GOES-16, GOES-17, CERES/Terra and CERES/Aqua have overlap in time and that the overlap is large enough to compare all of them.

**Reviewer # 2**

There is no high-resolution TOA SW flux dataset ground truth dataset, agreed, that is the motivation for this product. The CERES dataset provides observed instantaneous SW fluxes at the 20-km nominal resolution. Linear interpolating the CERES footprint center fluxes across the ABI pixels does not represent a valid 2-km flux field. Cloud edges are not distinct.
This implies that the ABI high-resolution TOA SW fluxes should be mapped into the CERES footprint for validation. Or into lower resolution latitude and longitude grid such as performed by Akkermans, T.; Clerbaux, N. Narrowband-to-Broadband Conversions for Top-of-Atmosphere Reflectance from the Advanced Very High Resolution Radiometer (AVHRR). Remote Sens. 2020, 12, 305. https://doi.org/10.3390/rs12020305
Note they also stratify the validation results by IGBP type. The important part of the validation is determining whether the algorithm is not adding an overall bias to the TOA SW fluxes, while trying to reduce the RMS error. The instantaneous RMS error is a function of spatial scale. They also validate a several year's worth of TOA fluxes

Response
For the re-mapping, we adopted the ESMF re-gridding package. The detailed information can be found at:

http://earthsystemmodeling.org/regrid/

For an ideal situation, the ABI high-resolution TOA SW fluxes should be mapped into the CERES footprint for validation as suggested by the Reviewer. However, there are reasons that make it difficult to do so. For example, the case 12/26/2019 UTC 19. There could be more than 18000 pixels in a single swath of the SSF if constrained to the region of U.S. Different pixels have different times. Neglecting the seconds, there are still more than 30 mins differences (this changes case by case) between the first pixel and the one at the end and this brings up a time matching issue. But if remapping the SSF to ABI, we can set up a unique time for ABI (ABI is at 5 min intervals) and then constrain the region and the time range of SSF.

Both remapping the ABI to SSF and re-mapping SSF to the ABI bring up spatial matching errors as recognized by the scientific community. In Figure 10, we show the SSF before re-gridding (Figs 10 (a) & (b)) and after re-gridding (Figs. 10 (c) and (d)). As seen, the fluxes after re-mapping CERES SSF to the ABI resolution resemble well the original CERES. A case of reverse mapping is shown in the Supplement Section and indeed as the Reviewer suggested, it reduces the edge effects. Another consideration is the computational efficiency of re-mapping the curvilinear tripolar grid to unconstructed grid. For large arrays, it is more efficient to remap the unconstructed grid to the curvilinear tripolar grid.

[Figure]

Part of Figure 10. (a) All sky TOA SW from CERES SSF/Aqua, (b) CERES SSF/Terra, (c) re-gridded CERES SSF/Aqua, (d) re-gridded CERES SSF/Terra.

We have done one case of remapping the ABI to CERES_SSF as suggested, and the edges do improve. There are additional consideration in selecting the direction of re-mapping. This Figure will be put in the Supplements.

**Remapping 2-km ABI flux to CERES_SSF scale (20 km)**

[Figure]

Figure S1. Top Left: Mean ABI Flux on 12/26/2019 UTC 19:00 from GOES-16 re-gridded to CERES SSF (20km)/Aqua domain; Top Right: Difference between re-gridded ABI Flux and CERES SSF/Aqua; Bottom: frequency distribution of the differences (bottom).

**Reviewer # 2**

The bin/channel regression rely on RTM results that have varying PW and ozone concentration. The PW water above the cloud or clear-sky surface is necessary to predict the NIR water vapor absorption, since none of the ABI band used are located inside absorption bands. It is not clear to me how current algorithm accounts for NIR water vapor absorption? This was unclear in section 2.

Response

[Figure]

ABI channel 4 (1.37um) is in the $H_2O$ absorption band, although not at the center.

[Figure]

Figure.        Sensitivity of channel 4 and 6 narrowband albedos and the broadband albedo to changes in column water vapor amount for a clear-sky ocean scene with fixed values of aerosol optical depth (AOD=0.13), ozone amount (O3=0.318 atm-cm) and cosine of solar zenith angle of 0.5. The ratio [A(i+1)-A(i)]/[H2O(i+1)-H2O(i)] is shown as a function of total column H2O. Here i runs from 1 to 5 and represents the five different H2O amounts (H2O) and the corresponding albedos (A).

Reviewer # 2
Could the MODIS 2-week surface band BRDF be used in MODTRAN to update the predefined MODTRAN BRDFs? The MODIS BRDF product could be used to account for regional and seasonal variability.

**Response**
Yes, in principle it can be done but we have worked with the MODTRAN built-in BRDFs.

**Specific Comments:**

**Reviewer # 2**
Line 38 this paragraph seems out of place. Unless this study was used to for ABI channel selection it does not seem relevant.

**Response**
We believe that this paragraph gives an idea of the difficulties faced performing the study described in our paper. We would like to keep it.

**Reviewer # 2**
Line 45. It would be beneficial for the reader to briefly outline the whole algorithm. To discuss both the indirect path and I am assuming a direct path. Perhaps to provide how this algorithm was developed and if it is used in any historical products.

**Response**
The manuscript describes estimation of the broadband TOA reflected flux from ABI observations. In this respect, the entire algorithm developed for this is fully described in the manuscript. While the work discussed here was performed in support of the development of the NOAA STAR Shortwave Radiation Budget (SRB) algorithm for estimating reflected shortwave fluxes at TOA (RSR) and downward shortwave fluxes at the surface (DSR) from ABI observations, the entire discussion could be entirely detached from the STAR algorithm.

Depending on the type of information available, there are two approaches to estimate RSR. When a full description of the atmosphere (gas amounts, spectral optical depth of aerosols and clouds, cloud phase, etc.) and the surface (spectral surface reflectance) are available one could input these into a radiative transfer model and calculate RSR. This is referred to as the "direct path" approach. An alternative to the direct path is to estimate a broadband reflectance from the narrowband ABI reflectance applying a narrowband to broadband conversion, and then an angular distribution model (ADM) to estimate a broadband albedo, from which RSR is calculated. Here this is referred to as the "indirect path" approach.
The NOAA STAR SRB algorithm implements both approaches. Details of the implementation are given in the Algorithm Theoretical Basis Document (ATBD) for Downward Shortwave Radiation (Surface), and Reflected Shortwave Radiation (TOA) (https://www.star.nesdis.noaa.gov/goesr/documents/ATBDs/Baseline/ATBD_GOES-R_Shortwave%20Radiation_3.1_Nov2018.pdf) and Laszlo et al (2008, 2020). The direct path is based on the NASA Clouds and the Earth's Radiant Energy System (CERES)/Surface and Atmospheric Radiation Budget (SARB) algorithm (Charlock and Alberta, 1996) algorithm, while the indirect path has its heritage in the in the GOES surface and insolation product (GSIP, Pinker et al., 2002) and build on the method described in Pinker and Laszlo (1992). The indirect path algorithm was also used to generate a product in the National Aeronautics and Space Administration (NASA)/Global Energy and Water Exchange (GEWEX) Surface Radiation Budget

project (Whitlock et al., 1995; Stackhouse et al., 2011). Even though it is implemented, the NOAA STAR SRB algorithm does not currently use the direct path since the ABI surface albedo product needed to run it is not yet available operationally.

**Reviewer # 2**
Line 64 Does ground refer to truth dataset or to actual ground observations, since in the summary mentions "ground truth"

**Response**
"ground truth" refers to the CERES TOA reflected flux. It is used as a reference data in the evaluation of the ABI TOA reflected flux.

**Reviewer # 2**
Line 95 Are the Kato and Loeb snow ADMs used as part of the CERES Ed2 ADMs?
Kato, S., and N. G. Loeb (2005), Top-of-atmosphere shortwave broadband observed radiance and estimated irradiance over polar regions from Clouds and the Earth's Radiant Energy System (CERES) instruments on Terra, J. Geophys. Res., 110, D07202, doi:10.1029/2004JD005308

**Response**
Yes, as mentioned in Table 2. The above reference is in the latest version of the paper.

**Reviewer # 2**

Line 97 The Niu and Pinker are theoretical simulations, how do they translate to observation numbers in Eq. 3?

**Response**
The comments of this Reviewer are well taken. We were also concerned about the issues he/she has raised. We have done numerous experiments to understand the sources of differences between the theoretical and CERES ADMs to convince ourselves that the synthesis of the two is sound, even if the two approaches are not identical. In **Figure 1** the patterns of bi-directional correction differences for desert under clear-sky from MODTRAN simulations and CERES observations are illustrated. Largest difference occurs for higher VZAs. While inaccuracies in the specific surface spectral reflectance used in the simulations may contribute to the differences, our experiments show that they are most likely due to differences in sampling frequency of observations at high VZAs. A hybrid approach is applied that hopefully is compensating for the uneven-sampling in the two methods.

[Figure]

Correction Factor

*Figure 1.        Bi-directional correction factors at SZA 63.2° over desert for clear-sky
Left: Simulations; Right: CERES observations (Bright Desert)*

Before undertaking the simulations, we had to develop a method to reconcile different scene types and angular binning of the CERES and simulated ADMs and a weighting function to combine the two data sources. CERES-TRMM clear-sky ADM classification by surface types does not fully match the IGBP surface classification. In the simulations, the 12 IGBP surface classifications are used. For clear sky, there are 8 surface types in CERES ADMs. An effort was made to combine the corresponding CERES ADMs and simulated ADMs based on IGBP scene classifications to generate new synthesized ADMs for 12 IGBP surface types. The cloud classification in CERES ADMs is based on Cloud Optical Depth (COD) and cloud phase (water cloud, ice cloud) over ocean, low-mod tree/shrub, mod-high tree/shrub, desert, and snow/ice.

For clear sky, the synthesized ADMs are generated from a combination of simulated and CERES bi-directional correction factors based on IGBP surface classifications for each angular bin by weighting as presented in the manuscript. For example, CERES Low-Mod Tree/Shrub ADMs are grouped from observations of the following three IGBP surface scenes: Savannas, Grassland, and Crops/Mosaic (Loeb et al., 2003). The difference in the bi-directional correction factors between the combined and CERES ADMs for Savannas is shown in **Figure 2.** At lower viewing zenith angles the percentage of differences is mostly within +/- 10% but the differences are much larger at higher viewing zenith angles.

[Figure]

*Figure 2.      Distribution patterns of the difference of the bi-directional correction factor between combined ADMs and CERES ADMs for Savannas over clear sky at Solar Zenith Angle of 70-80°:*
*Left:    Difference (Combined ADMs – CERES ADMs)*
*Right:  Percentage of Difference (Difference/CERES ADMs)*

At an early stage of this work when ABI observations were not yet available, we have tested the approach with SEVIRI observations. The following Table (Niu and Pinker, 2011) illustrates that using the hybrid approach results in better agreement with CERES compared to what was achievable with CERS ADMs alone.

Table 7. Evaluations of July 2004 monthly mean TOA upward SW flux estimates as driven with SEVIRI observations when using CERES ADMs or synthesized ADMs, against CERES observations (SRBAVG product).

| | BIAS (W m$^{-2}$) | | RMSE (W m$^{-2}$) | |
|---|---|---|---|---|
| Statistical results | CERES ADMs | Synthesized ADMs | CERES ADMs | Synthesized ADMs |
| Clear sky (7801 samples) | 8.7 | 4.6 | 7.1 | 6.5 |
| All sky (8128 samples) | −3.1 | −2.7 | 8.2 | 6.3 |

As mentioned in our manuscript, we have originally prepared two papers. The first one summarized the early results with proxy observations like SEVIRI, GERB, MODIS etc. where some of these issues are explained in detail. Due to concern that the early material may not be any more of interest to the readers, we have focused in the second paper on ABI using the latest versions of GOES-16 and 17 data.

**Reviewer # 2**
Line 131. I do not see how the Fig. 4 comparison adds value to the paper. The profiles were selected to get a sampling of the diversity of atmospheric profiles found on Earth.

**Response**
From past experience, we are aware that some Reviewers like to see how the sampling was done so we would like to keep Figure 4.

**Reviewer # 2**
Line 237 Is the Matlab stepwise fit used in the algorithm? If not this should sentence should be left out because it adds confusion.

**Response**
We have derived coefficients of regression using a constrained least-square curve fitting method of Matlab, "lsqnonneg", which can solve a linear or nonlinear least-squares (data-fitting) problem

and produce non-negative coefficients. Non-negative coefficients avoid generating negative TOA flux, which is not physically valid.

To ensure that information from all channels is used and avoid the complex cross-correlation problem, it was opted to generate Narrow to Broad (NTB) coefficients for each ABI channel separately. These channel specific NTB coefficients are applied to each channel to convert ABI narrow-band reflectance to extended band. The final broad-band TOA reflectance is taken as the weighted sum of all 6-channel specific broad-band reflectance. The logic behind this approach is the assumption that the narrow-band reflectance from each channel is a good representative for a limited spectral region centered around the channel and the total spectral reflectance is dominated by the spectral regions that contains the most solar energy

**Reviewer # 2**
Fig. 8 Could the spectral boundaries or band edges for each ABI band also be shown in Fig. 8. This way the reader can see the spectral range radiance that is predicted based on a single ABI band.

**Response**
The band edge values are listed in the text.

**Reviewer # 2**
Line 260 could the band edges be given in μm in the text also.

**Response**
The band edge values are listed now also in μm. They are:
0.2001    0.5341    0.7584    1.0845    1.4680    1.8896    4.0000 μm

**Reviewer # 2**
Section 2.6 Which channel takes into account the bulk of the NIR water vapor absorption?

**Response**
Channel 6 (2.25 μm) and to some extent channel 4 (1.37 μm).

**Reviewer # 2**
Line 264 Figure 9 is spelled out, whereas Fig. 8 is not on line 261

**Response**
Corrected. Thank you.

**Reviewer # 2**
Line 266. I would agree that along the cloud edges there would be large differences between ABI and CERES TOA fluxes. These large differences would occur even if there were a perfect

algorithm. However, over large spatial domains the ABI and CERES fluxes should be similar.

**Response**
Indeed, it is so.

**Reviewer # 2**
Table 6 and 7 are not referenced in the text.

**Response**
Inserted now.

**Reviewer # 2**
Line 267 It would be nice to have statistics for Figure 9 similar to what is in Table 7. I do not see 2017/11/25, 17:57Z Fig 9 statistics with the 2019 statistics in Table 7.

**Response**
We have added it now to Figure 9.

[Figure]

**Reviewer # 2**
Line 276. This is where Table 6 should be referenced to identify the CODC product

**Response**

We have now referenced it in the text.

**Reviewer # 2**
Line 283. The authors should use the CERES SSF Level 2 data, rather than CERES FlashFLUX footprint fluxes. As mentioned in the text, that FlashFLUX does not use the most up to date CERES instrument calibration coefficients. The CERES SSF product is available within 3-

months of real-time.

**Response**
Indeed, this was done as explained before.

**Reviewer # 2**
What is limiting the number of validation match ups? Is the issue that your computing resources have limited computer storage that downloading all of the required datasets for ABI pixel level fluxes and comparisons with CERES is not possible after real-time when these products are no longer available at CLASS?

**Response**
The 2-km Pixel-level ABI fluxes are not yet available at CLASS, so these had to be generated and stored locally. The other one is the matching.

**Reviewer # 2**
Line 304. The CERES footprint data has a resolution of 20-km at nadir, while the ABI pixel has 2-km resolution. By linear interpolating spatially the CERES fluxes across the ABI pixel does not properly distribute spatially the CERES flux observation (by not preserving cloud edges) and I would not consider that a truth dataset, since it does not represent the observed 2-km fluxes, It would be better to map the ABI pixels into the CERES footprint to validate the NTB algorithm. A CERES footprint at 60° view angle (near the scan edge) has a 40-km extent encompassing over 400 ABI pixels at nadir. Even better would be to evaluate the ABI product regionally, say for 1°

**Response**
For the re-mapping, we adopted the ESMF re-gridding package. The detailed information can be found at:
http://earthsystemmodeling.org/regrid/
For an ideal situation, the ABI high-resolution TOA SW fluxes should be mapped into the CERES footprint for validation as suggested by the Reviewer. However, there are reasons that make it difficult to do so. For example, the case 12/26/2019 UTC 19. There could be more than 18000 pixels in a single swath of the SSF if constrained to the region of U.S. Different pixels have different times. Neglecting the seconds, there are still more than 30 mins differences (this changes case by case) between the first pixel and the one at the end and this brings up a time matching time issue. But if remapping the SSF to ABI, we can set up a unique time for ABI (ABI is at 5min intervals) and then constrain the region and the time range of SSF.

Both remapping the ABI to SSF and remapping SSF to the ABI bring up spatial matching errors as recognized by the scientific community. In Figure 10, we show the SSF before re-gridding (Figs 10 (a) & (b)) and after re-gridding (Figs. 10 (c) and (d)). As seen, the fluxes after re-mapping CERES SSF to the ABI resolution resemble well the reverse re-mapping. Another consideration

is the computational efficiency of re-mapping the curvilinear tripolar grid to unconstructed grid. For large arrays, it is more efficient to remap the unconstructed grid to the curvilinear tripolar grid.

We have done remapping to 1° s suggested. Here is the result. Will add it to Supplement.

**Remapping ABI flux to 1°**

[Figure]

Figure S2. CERES_SSF1deg/Aqua (top left), ABI Flux from GOES16 re-gridded to 1°, the difference between re-gridded ABI Flux and CERES_SSF1deg/Aqua (bottom left) and the frequency distribution of the differences (bottom right).

**Reviewer # 2**
Fig 10 caption missing (e)

**Response**
We have re-drafted Figures 10-13 using CERES SSF (20 km) to compare with ABI. Legends have been adjusted accordingly.

[Figure]

Figure 10. (a) All sky TOA SW from CERES_SSF/Aqua, (b) CERES_SSF/Terra, (c) re-gridded
CERES_SSF/Aqua, (d) re-gridded CERES_SSF/Terra, (e) GOES-16 and (f) GOES-
17 on 12/26/2019 at UTC 19:36.

[Figure]

Figure 11. (a) Frequency distribution of all-sky TOA SW differences between ABI on GOES-16 and CERES and (b) ABI on GOES-17 and CERES_SSF using Aqua (c) and (d) as above for Terra. All observations were used (clear and cloudy) on 12/26/2019 at UTC 19:36.

[Figure]

Figure 12. Same as Figure 11 for clear sky TOA SW differences.

[Figure]

Figure 13. Same as Figure 11 but for cloudy TOA SW differences.

**Reviewer # 2**

Line 326 Based on Table 6, the ABI radiances, aerosols, cloud mask, phase and optical depth are used as inputs. For clear-sky the surface spectral reflectance is based on 12 IGBP types, and 4 types for cloudy types. How is the pixel level above surface or cloud top amount to account for NIR atmospheric water vapor absorption. A lot of effort was used to define atmospheric profiles, I would assume this would be based on the ABI channel radiances. My other concern is that the

0.86 vegetation reflection is a function of season and region, in winter the leaves have fallen off the trees, where as in summer the trees have leaves. By simply relying on IGBP type does not account for the seasonal vegetation reflection.

**Response**

We have noted the problem of seasonality. It is not trivial to incorporate it.

**Reviewer # 2**

Line 346 and line 33. Given that the ABI sampling is less than 15 minutes. The 7.5 minute difference is very small. Once the SW fluxes are compared at the footprint or regional scales the time difference will not make much of a difference in the bias. All Terra and Aqua overpasses should be matched for well sampled validation results. The following paper Fig. 2 shows that the time difference does not dramatically increase the matching noise
B. A. Wielicki, D. R. Doelling, D. F. Young, N. G. Loeb, D. P. Garber and D. G. MacDonnell, "Climate Quality Broadband and Narrowband Solar Reflected Radiance Calibration Between Sensors in Orbit," IGARSS 2008 - 2008 IEEE

**Response**
Thank you for pointing this out to us. We have now referenced this finding of Wielicki et al. Indeed, for large scale the 15 min difference does not show up at cloud edge/.

**Reviewer # 2**
Line 348 I agree that seasonal/regional variation of the NIR vegetation reflection must be taken into account.

**Response**
Thank you.

**Reviewer # 2**
Line 358 The CERES edition 4 ADMs also rely on NDVI, which accounts for changes in the vegetation NIR reflectance. The CERES edition 2 relies on surface types only.
Su, W., Corbett, J., Eitzen, Z., and Liang, L.: Next-generation angular distribution models for top-of-atmosphere radiative flux calculation from CERES instruments: methodology, Atmos. Meas. Tech., 8, 611–632, https://doi.org/10.5194/amt-8-611-2015, 2015

**Response**
We have used CERES edition 2 which relies on surface types only.

**Reviewer # 2**

Line 38- what is the source of the open shrub, desert, woody savanna and grassland spectral albedos? Are these TOA albedos?

**Response**

They are from the MODTRAN model and are spectral surface albedos.

**Reviewer # 2**

Line 397. I agree there is no truth dataset for 2km resolution BB fluxes. That is the reason why this dataset is being produced. In order to perform a fair comparison, the high-resolution ABI pixels fluxes must be mapped into the CERES footprint, or both reduced to a 100-km region in order to track the ABI and CERES over the record.

**Response**

We have responded to this comment in response to Reviewer # 2 comment to line 304.

Line 405. "transformation of narrowband quantities into broadband ones" This sentence is ambiguous.

**Response**

Not clear to us what is wrong here

**Reviewer # 2**

Line 414. What is this sentence trying to say? "The process of preparing for the usefulness of a new satellite sensor needs to be done in advance, the final configuration of the instrument becomes known at a much later stage." This was not addressed in the paper

**Response**

Has been removed.

**Reviewer # 2**

Line 416 What is this sentence trying to say? "As such, the evaluation of the new algorithms is in a fluid stage for a long time." Usually there is an initial release and as the algorithms improve incrementally while the version number is updated over time. For example MODIS L1b C6.1 dataset is currently available and C7 is being developed and tested.

**Response**

We believe that the decision when to release preliminary results depends on the situation at hand. As long as it is not possible to have some evaluation of the product, it may be counter-productive to release the data.

**Reviewer # 2**

Line 417 This sentence is confusing. "Agreement or disagreement with know "ground truth" is not fully informative on the performance of the new algorithms to estimate desired geophysical parameters." Are you talking about compensating errors?

**Response**

The sentence is not clear. Is now reworded.

**Reviewer # 2**

Line 420 reliable cloud screening and cloud properties. What about non-retrieved cloud properties from cloud mask identified pixels? What about optically very thin clouds where the surface contributes to the TOA reflectance.

**Response**

For such cases, no retrievals were done.

**Reviewer # 2**

Line 421 The CERES SSF L2 Edition 4 product SW fluxes has been available prior to ABI and have not gone through any major revisions. The SSF1deg fluxes have been used to monitor global and regional SW flux variability over time. On the other hand the FlashFLUX has undergone revisions.

**Response**

Indeed, that is so.

**Reviewer # 2**

What is the application of high-resolution ABI TOA SW fluxes, that the low resolution CERES fluxes cannot fulfill? For the application, what is the required SW TOA accuracy?

**Response**

That resolution is needed for implementing the SW algorithm at NOAA/STAR. If other users are interested in the product, it is up to them to determine if the accuracy as established against CERES (the only game in town) is sufficient.

Line 429 If the ABI aerosol algorithm does not ever reach stability in the future, will the TOA SW product ever be released?

**Response**

Not clear how these two issues are related. The TOA SW fluxes at this stage do not depend on the aerosol algorithm. To incorporate the real time aerosol product into the TOA algorithms is a future project.

---

## Author Comment (AC4)

**Comments to the author**:
Dear authors,
thank you for submitting your revisions. My apologies for the delayed processing on my end; I was on field campaign travel for a month. I need to run this by one of the reviewers. I can see that some changes were made, and I lean towards accepting the paper. However, I'd like to better understand some elements of the exchange, including:

Reviewer
If the GOES SW TOA flux product is being produced by NOAA it should be cited. If it is not, then it should also be stated in the text. If this is an algorithm paper of a potential NOAA product that is in development that should be clearly stated.

Authors
Answered to previous comment. It is not a potential product. This is an existing product.

Editor:
Here, it is not clear to me how the paper can be, at the same time, the description of a development effort for products that are posted online, but also an existing product. If this is the introduction of a new product, then the current paper is the first algorithm paper, which would be fine. If a previous paper describing the algorithm exists (which I doubt), then the response does not make sense, in my opinion. Can you clarify? I might be misunderstanding something.

Reviewer
3.1 Satellite data for GOES-16 and GOES17: datasets are used in papers I expect the product name, version number and location should be given. I find section 3.1 completely lacking in this regard. First of all, I searched for https://www.bou.class.noaa.gov/ and the site could not be found. I do not know if this is the GOES L1b radiance data, since the product name was not given in the text.

Editor:
I agree with the reviewer here that section 3.1 is too short, considering that there is no other paper to go to for more explanation. The table and the caption that is used here, along with the link, are insufficient, in my mind, and some text needs to be written around the product, UNLESS another paper can be cited. Here again, I might be mis-understanding something, but again, a short paragraph on the data description does not sound sufficient to me. I am glad that the reviewer brought this up, and I am sorry to keep insisting on taking the reviewer's feedback to heart.

Reviewer
3.2 Reference data from CERES [...]

Authors
CERES SSF version 4a and FlashFlux version 3c data were used

Editor:
I can see that changes were made in the manuscript text, but please also state in the

response to the reviewers which changes were made in the manuscript to address this particular comment as is common practice.

Reviewer
I looked at the ESMF re-gridding web site, there are multiple grid type options. Could the gridding algorithm just be simply detailed in the text.[...]

Authors:
The ESMF re-gridding program is a complicated package. [...] We felt that an interested user will have to go back to that package and not to rely on a brief summary.

Editor:
I generally agree that information on data attributes (such as the grid they are defined on) that is available elsewhere does not have to be repeated in a publication. However, some of the information you provided in your response to the reviewer should be included in the paper, at least at a superficial level so that the interested reader knows where to go. Also, I am missing a response to the reviewer's comment regarding surface types.

Reviewer
In the abstract the last sentence states: A satisfactory agreement between the fluxes was observed for both clear and cloudy conditions and possible reasons for differences have been identified." Satisfactory agreement is a relative term. I believe that the authors need to describe who their users are and that the level of agreement is sufficient for their applications.

Editor:
I agree with the response that it is impossible to know the users ahead of time. However, I also agree with the reviewer that "satisfactory" is too relative of a term. A quantitative (rather than qualitative as currently provided) statement would be more befitting of this statement in the abstract.

Since I posted my comments in the interactive discussion, can you please post your response (with any additional edits given my comments above) in direct reference to EC1 by clicking on the "reply" button? That way the exchange is public and part of the record.

05/05/2022

Dear Editor:

We will respond to each comment as it appears in your communication. However, we have a feeling that something went amiss here. Possibly, the Reviewer was looking at an older version of the manuscript since some of the issues raised were already responded to. I will illustrate with some examples:

**Example # 1**

**The Reviewer writes:**

3.2 Reference data from CERES [...]

Authors

CERES SSF version 4a and FlashFlux version 3c data were used

Editor:
I can see that changes were made in the manuscript text, but please also state in the response to the reviewers which changes were made in the manuscript to address this particular comment as is common practice.

**Author Response**

Please go to:

amt-2021-289-ATC2.pdf Date: 04 Apr 2022, Status: File upload (AMT), Iteration: Minor revision, Finalized: Yes

**It is stated explicitly in the response to the Reviewer that:**

CERES SSF version 4a and FlashFlux version 3c data were used

**Example # 2**

Reviewer
In the abstract the last sentence states: A satisfactory agreement between the fluxes was observed for both clear and cloudy conditions and possible reasons for differences have been identified." Satisfactory agreement is a relative term. I believe that the authors need to describe who their users are and that the level of agreement is sufficient for their applications.

Editor:
I agree with the response that it is impossible to know the users ahead of time. However, I

also agree with the reviewer that "satisfactory" is too relative of a term. A quantitative (rather than qualitative as currently provided) statement would be more befitting of this statement in the abstract.

**Author Response**

There is no such statement in the latest version of the manuscript as submitted under:

amt-2021-289-ATC2.pdf Date: 04 Apr 2022, Status: File upload (AMT), Iteration: Minor revision, Finalized: Yes

**Now we will respond to each comment as it appears in your communication**

Reviewer
If the GOES SW TOA flux product is being produced by NOAA it should be cited. If it is not, then it should also be stated in the text. If this is an algorithm paper of a potential NOAA product that is in development that should be clearly stated.

Authors
Answered to previous comment. It is not a potential product. This is an existing product.

Editor:
Here, it is not clear to me how the paper can be, at the same time, the description of a development effort for products that are posted online, but also an existing product. If this is the introduction of a new product, then the current paper is the first algorithm paper, which would be fine. If a previous paper describing the algorithm exists (which I doubt), then the response does not make sense, in my opinion. Can you clarify? I might be misunderstanding something.

**Authors Response**

In our previous response we have stated the following:

**Authors**
The paper is about the development of methodology to derive TOA SW fluxes at NOAA STAR. This product is a starting point for deriving surface SW fluxes when using the "indirect approach". There is also a need to know how well the proposed methodology is working. Therefore, the evaluation of the methodology against best available estimates of TOA fluxes is an important element of the paper. The TOA reflected SW flux is produced together with the surface downward SW flux and archived at the NOAA Comprehensive Large Array-data Stewardship System (CLASS) at avl.class.noaa.gov as archived at the NOAA Comprehensive Large Array-data Stewardship System (CLASS) at avl.class.noaa.gov. It is an end-product just like the surface flux. Since the TOA and surface fluxes

are generated together in the same process by the same algorithm the product/algorithm version numbers are the same.
It is an intermediate product and as such, versions have the same labeling as the final product, namely, the surface SW fluxes.
The method for estimating the TOA broadband albedo developed in the effort documented in the paper has been applied in an algorithm that is used by NOAA to operationally generate the level 2 (L2) reflected shortwave radiation at TOA product since the launch of GOES 16 in November 2016. This product is archived and can be freely downloaded from, the NOAA Comprehensive Large Array-data Stewardship System (CLASS) at avl.class.noaa.gov, in the "GOES-R Series ABI Products (GRABIPRD)" category under the name of "Reflected Shortwave
Radiation: TOA". The algorithm/product version number is version 1, revision 0 (v01r00). For the ABI clear-sky mask (cloud mask) and cloud optical depth are also v01r00.

Since the above text seems to be lacking in clarity, we are smoothing the text so hopefully there is no ambiguity. Here is the new version:

This is a first paper that describes the development of a methodology to derive TOA SW fluxes from the Advanced Baseline Imager onboard the NOAA GOES-R series of geostationary satellites that are used at NOAA STAR as a starting point for deriving surface SW fluxes. To find out how the methodology is working evaluation of the methodology against best available estimates of TOA fluxes was also done. The TOA reflected SW flux is produced at NOAA together with the surface downward SW flux and is archived at the NOAA Comprehensive Large Array-data Stewardship System (CLASS) at avl.class.noaa.gov. While the TOA reflected SW flux is a product on its own right, it is also a prerequisite to deriving the SW surface flux; as such, versions for TOA and surface have the same labeling.

We will now add this text to Introduction

Reviewer
3.1 Satellite data for GOES-16 and GOES17: datasets are used in papers I expect the product name, version number and location should be given. I find section 3.1 completely lacking in this regard. First of all, I searched for https://www.bou.class.noaa.gov/ and the site could not be found. I do not know if this is the GOES L1b radiance data, since the product name was not given in the text.

Editor:
I agree with the reviewer here that section 3.1 is too short, considering that there is no other paper to go to for more explanation. The table and the caption that is used here, along with the link, are insufficient, in my mind, and some text needs to be written around the product, UNLESS another paper can be cited. Here again, I might be mis-understanding something, but again, a short paragraph on the data description does not sound sufficient to me. I am glad that the reviewer brought this up, and I am sorry to keep insisting on taking the reviewer's feedback to heart.

Authors Response

The referenced site was:

www. avl.class.noaa.gov

As such, when the search was done for

https://www.bou.class.noaa.gov/

nothing was found.

Text was added to the manuscript.

The Advanced Baseline Imager (ABI) data used (Table 6) were downloaded from the NOAA Comprehensive Large Array-Data Stewardship System (CLASS) at https://www.avl.class.noaa.gov/saa/products/welcome . Both level 1b (L1b) and level 2 (L2) data were used. These can be found by searching the CLASS site by selecting "GOES-R Series ABI Products GRABIPRD (partially restricted L1b and L2+ Data Products)".The L1b data included the radiances (RadC) in files "OR_ABI-L1b-RadC-MmCnn_G1SS_stime_etime_ctime, where "m", "nn" and "SS" indicate the ABI scan mode, channel number (01-06) and satellite identification number (16 or 17), respectively. "stime", and "etime" are the start and end dates and times of the scan, "ctime" is the date and time the file was created on. The ABI L2 product used were the clear-sky mask, cloud top phase, cloud optical depth. The names of these files are constructed similarly to the L1b radiance files, except that the radiance product name RadC is replaced by ACMC, ACTPC, CODC and AODC, respectively, and the reference to the channel number is omitted. For example, GOES-16 with ABI operating in scan mode 6 in the CONUS domain, the name of the clear-sky mask file is OR_ABI-L2-ACMC-M6_G16_ stime_etime_ctime. (In the product names above the letter C indicates the CONUS domain.)

The clear-sky mask product consists of a binary cloud mask identifying pixels as clear, probably clear, cloudy or probably cloudy. The cloud top phase product provides cloud classification identification information for each pixel. The cloud phase categories are clear sky, liquid water, super cooled liquid water, mixed phase, ice, and unknown. The cloud optical depth product gives the optical thickness along an atmospheric column for each pixel. All products have a nominal sub-satellite spatial resolution of 2 km.

Reviewer
3.2 Reference data from CERES [...]
Authors

CERES SSF version 4a and FlashFlux version 3c data were used

**Authors Response**

Editor:
I can see that changes were made in the manuscript text, but please also state in the response to the reviewers which changes were made in the manuscript to address this particular comment as is common practice.

**Authors Response**

As explained in Example # 1, this statement is in our response as can be seen in: amt-2021-289-ATC2.pdf Date: 04 Apr 2022, Status: File upload (AMT), Iteration: Minor revision, Finalized: Yes

Reviewer
I looked at the ESMF re-gridding web site, there are multiple grid type options. Could the gridding algorithm just be simply detailed in the text.[...]

Authors:
The ESMF re-gridding program is a complicated package. [...] We felt that an interested user will have to go back to that package and not to rely on a brief summary.

Editor:
I generally agree that information on data attributes (such as the grid they are defined on) that is available elsewhere does not have to be repeated in a publication. However, some of the information you provided in your response to the reviewer should be included in the paper, at least at a superficial level so that the interested reader knows where to go. Also, I am missing a response to the reviewer's comment regarding surface types.

**Authors Response**

Such information is provided. It reads:

3.3 Data preparation

290 291 For the re-mapping, we adopted the ESMF re-gridding package. The detailed information can be found
292 at: http://earthsystemmodeling.org/regrid/
293 For an ideal situation, the ABI high-resolution TOA SW fluxes should be mapped into the CERES
294 foot-print for validation. However, there are reasons that make it difficult to do so. There can be more than
295 18000 pixels in a single swath of the SSF, when constrained to U.S. Different pixels have different times.
296 Neglecting the seconds, there are still more than 30 mins differences (this changes case by case) between
297 the first pixel and the one at the end and this brings up a time matching issue. By remapping the SSF to 13

298 ABI, we can set up a unique time for ABI (ABI is at 5 min intervals) and then constrain the region and
299 the time range of SSF.
300 Both re-mapping the ABI to SSF and remapping SSF to the ABI bring up spatial matching errors as
301 recognized by the scientific community (Rilee and Kuo, 2018; Ragulapati et al., 2021). In Fig. 11, we
302 show the SSF before re-gridding (Figs 11 (a) & (b)) and after re-gridding (Figs. 11 (c) and (d)). The
303 fluxes after re-mapping CERES SSF to the ABI resolution resemble well the original structure. Another
304 consideration is the computational efficiency of re-mapping the curvilinear tripolar grid to unconstructed
305 grid. For large arrays, it is more efficient to remap the unconstructed grid to the curvilinear tripolar grid.

Perhaps, the comment:

The detailed information can be found at: http://earthsystemmodeling.org/regrid/

that appears up-front, should have been placed at the end of the section. We have done so now. Should be clearer. Thank you.

Reviewer
In the abstract the last sentence states: A satisfactory agreement between the fluxes was observed for both clear and cloudy conditions and possible reasons for differences have been identified." Satisfactory agreement is a relative term. I believe that the authors need to describe who their users are and that the level of agreement is sufficient for their applications.

Editor:
I agree with the response that it is impossible to know the users ahead of time. However, I also agree with the reviewer that "satisfactory" is too relative of a term. A quantitative (rather than qualitative as currently provided) statement would be more befitting of this statement in the abstract.

**Authors Response**

We have checked the latest version of the manuscript that was submitted:

amt-2021-289-manuscript-version4.pdf Date: 04 Apr 2022, Status: File upload (AMT), Iteration: Minor revision, Finalized: Yes

There is no such statement in the Abstract. Possibly, the Reviewer was looking at an earlier version of the manuscript.

Since I posted my comments in the interactive discussion, can you please post your response (with any additional edits given my comments above) in direct reference to EC1 by clicking on the "reply" button? That way the exchange is public and part of the record.

Will do. Thank you.
Thank you,

---

## Author Response (AR2)

Comment for manuscript amt-2021-289 on behalf of one of the reviewers.

Dear authors,

one of the reviewers communicated with me and sent some follow-up comments regarding the revised version of your manuscript. I post them here in the public discussion because they seem appropriate for the manuscript's public record. I encourage you to post a point-by-point response when making edits for the final version of the paper.
Thank you,
Sebastian Schmidt (editor)

Comment by reviewer in response to the revised version, and also in response to AC3 (https://doi.org/10.5194/amt-2021-289-AC3)

**Response**

**Reviewer**
The authors could have done more to address my initial feedback, addressing the following clarification comments would be greatly appreciated.
It is still unclear to me if the paper is a validation paper of the NOAA STAR TOA SW flux product and if so the dataset and version number should be properly cited.

**Authors**
The paper is about the development of methodology to derive TOA SW fluxes at NOAA STAR. This product is a starting point for deriving surface SW fluxes when using the "indirect approach". There is also a need to know how well the proposed methodology is working. Therefore, the evaluation of the methodology against best available estimates of TOA fluxes is an important element of the paper. The TOA reflected SW flux is produced together with the surface downward SW flux and archived at the NOAA Comprehensive Large Array-data Stewardship System (CLASS) at avl.class.noaa.gov as archived at the NOAA Comprehensive Large Array-data Stewardship System (CLASS) at avl.class.noaa.gov. It is an end-product just like the surface flux. Since the TOA and surface fluxes are generated together in the same process by the same algorithm the product/algorithm version numbers are the same.

It is an intermediate product and as such, versions have the same labeling as the final product, namely, the surface SW fluxes.
The method for estimating the TOA broadband albedo developed in the effort documented in the paper has been applied in an algorithm that is used by NOAA to operationally generate the level 2 (L2) reflected shortwave radiation at TOA product since the launch of GOES 16 in November 2016. This product is archived and can be freely downloaded from, the NOAA Comprehensive Large Array-data Stewardship System (CLASS) at avl.class.noaa.gov, in the "GOES-R Series ABI Products (GRABIPRD)" category under the name of "Reflected Shortwave

Radiation: TOA". The algorithm/product version number is version 1, revision 0 (v01r00). For the ABI clear-sky mask (cloud mask) and cloud optical depth are also v01r00.

**Reviewer**

If the GOES SW TOA flux product is being produced by NOAA it should be cited. If it is not, then it should also be stated in the text. If this is an algorithm paper of a potential NOAA product that is in development that should be clearly stated.

**Authors**

Answered to previous comment. It is not a potential product. This is an existing product.

**Reviewer**

3.1 Satellite data for GOES-16 and GOES17: datasets are used in papers I expect the product name, version number and location should be given. I find section 3.1 completely lacking in this regard. First of all, I searched for https://www.bou.class.noaa.gov/ and the site could not be found. I do not know if this is the GOES L1b radiance data, since the product name was not given in the text.

**Authors**

Web site addresses are frequently changed. Before submitting a paper or revisions, we always verify addresses we provide. At the time of submission, the links we provided did work. Please keep in mind that the review process of this manuscript took about **seven months,** increasing the chance for address change. The current address is:

https://www.avl.class.noaa.gov/saa/products/welcome

It has been updated now.
All the requested information is provided in Table 6. We felt that there is no need to repeat it in the text.

We suggest that the reader uses the keyword "class data noaa" to search with google.
And chose the "NOAA's Comprehensive Large Array-data Stewardship System".
Under that web-page, and in the search bar list it is clear that there is a "GOES-R Series ABI Products GRABIPRD (partially restricted L1b and L2+ Data Products)"

**Reviewer**

The text mentions that "The CODC data are not always available from CLASS".
Could the authors provide the name and version of the product of the cloud retrievals used in this study. Lastly the GOES based TOA flux dataset or product promoted in this paper is not cited in the paper.

**Authors**

This comment was placed now under Table 6. At the early stages of the CLASS archive, not all the needed information was available so it had to be imported from NOAA/STAR. Since, the archive was augmented.

**Reviewer**

3.2 Reference data from CERES: This section is completely confusing. Some of the figures were used from CERES SSF L2 and for fig. 9 the CERES FLASHflux level 2. Again, the edition numbers were not cited. I believe it was CERES SSF L2 Edition4 and FLASHflux Version 4A. This is extremely important if someone wanted to recreate the results in the future when the CERES project may have moved on to Edition 5.

**Authors**

CERES SSF version 4a and FlashFlux version 3c data were used

**Reviewer**

I was disappointed that only a few overpasses were validated in the paper and here is the response from the authors. "The ABI is at 5 min intervals. However, we want to compare four products simultaneously. It is hard to find cases when all of the GOES-16, GOES-17, CERES/Terra and CERES/Aqua have overlap in time and that the overlap is large enough to compare all of them." For me, there is no stipulation that they need to be validated simultaneously in order to have a robust validation matched dataset.

**Authors**

Indeed, it is difficult to convey in a paper of this type how much effort went into the evaluation during the entire process. As mentioned, at early stages, NOAA was downloading ("grabbing") data for short periods of time (about a week) for testing. These data were shared with us. Before the next download, such data are discarded to make space for a new set. It is not reasonable to ask for the version of such data. It also does not make sense to show results from these experiments since there is no way that interested parties could replicate such results. Therefore, we had to wait till there was a product that all parties can download (CLASS).
The Reviewer says: "For me, there is no stipulation that they need to be validated simultaneously in order to have a robust validation matched dataset."

This is a matter of opinion. From our experience, users that may be interested to use data from both satellites, want to know how the two satellites compare. Also, what if one satellite fails and after using data from GOES-16 they want to switch to GOES-17? Our approach was to anticipate such requests from users.

**Reviewer**

I looked at the ESMF re-gridding web site, there are multiple grid type options. Could the gridding algorithm just be simply detailed in the text.
The point of the paper is that the CERES and GOES surface types could be a factor. The Su et al. 2015 ADM type are more a function of NDVI over land and not strictly dependent on IGBP type and that NDVI allows for seasonal variability, whereas the GOES (this paper) has a static surface type categories not allowing for seasonal variation of interannual variability.

**Authors**

The ESMF re-gridding program is a complicated package. Information on grid type and remapping has been given in the original response. We have mentioned that "For large arrays, it is more efficient to remap the unconstructed grid to the curvilinear tri-polar grid."

The ESMF website gives a detailed description of how-to re-grid from one type to the other. We felt that an interested user will have to go back to that package and not to rely on a brief summary.

**Reviewer**

Line 389. The "ground truth", namely, the CERES observations are also undergoing adjustments and recalibration, is misleading. The CERES SSF L2 TOA flux observations have been using consistent algorithms and instrument calibration across a CERES edition (not FLASHflux). That is a new edition is reprocessed from the beginning of record with consistent algorithms and calibration. That is why citing datasets is so important.

**Authors**

There is no contradiction here. We agree that "a new edition is reprocessed from the beginning of record with consistent algorithms and calibration." When this happens, the older version is removed (in our experience). There is a possibility to encounter difficulty if results are based on a version that is no more accessible to the public. We now cite the data set used.

**Reviewer**

In the abstract the last sentence states: A satisfactory agreement between the fluxes was observed for both clear and cloudy conditions and possible reasons for differences have been identified." Satisfactory agreement is a relative term. I believe that the authors need to describe who their users are and that the level of agreement is sufficient for their applications.

**Authors**

We can add that the agreement is as shown in Table 6. The Reviewer writes: "I believe that the authors need to describe who their users are and that the level of agreement is sufficient for their applications." We do not know who the users are. It is up to the users to decide if the agreement reported is sufficient for their use.